# HERALD: A NATURAL LANGUAGE ANNOTATED LEAN 4 DATASET

**Guoxiong Gao**[1*]   **Yutong Wang**[2*]   **Jiedong Jiang**[1*]   **Qi Gao**[3*]   **Zihan Qin**[1]
**Tianyi Xu**[1]   **Bin Dong**[45†]

[1]Peking University   [2]National University of Singapore   [3]Center for Data Science, Peking University   [4]Beijing International Center for Mathematical Research and the New Cornerstone Science Laboratory, Peking University   [5]Center for Machine Learning Research, Peking University

`samggx@stu.pku.edu.cn`, `e1124791@u.nus.edu`
`emailboxofjjd@pku.edu.cn`, `qig@stu.pku.edu.cn`
`zihanqin@stu.pku.edu.cn`, `xutianyi@bicmr.pku.edu.cn`
`dongbin@math.pku.edu.cn`

## ABSTRACT

Verifiable formal languages like Lean have profoundly impacted mathematical reasoning, particularly through the use of large language models (LLMs) for automated reasoning. A significant challenge in training LLMs for these formal languages is the lack of parallel datasets that align natural language with formal language proofs. To address this challenge, this paper introduces a novel framework for translating the Mathlib4 corpus (a unified library of mathematics in formal language Lean 4) into natural language. Building upon this, we employ a dual augmentation strategy that combines tactic-based and informal-based approaches, leveraging the Lean-jixia system, a Lean 4 analyzer. We present the results of this pipeline on Mathlib4 as Herald (**Hie**rarchy and **R**etrieval-based Tr**a**nslated **L**ean **D**ataset). We also propose the Herald Translator, which is fine-tuned on Herald. Herald translator achieves a 96.7% accuracy (Pass@128) on formalizing statements in the miniF2F-test and a 23.5% accuracy on our internal graduate-level textbook dataset, outperforming InternLM2-Math-Plus-7B (73.0% and 7.5%) and TheoremLlama (50.1% and 4.0%). Furthermore, we propose a section-level translation framework for real-world applications. As a direct application of Herald translator, we have successfully translated a template section in the Stack project, marking a notable progress in the automatic formalization of graduate-level mathematical literature. Our model, along with the datasets, are open-sourced to the public.[1]

Keywords: Lean 4, Autoformalizing, LLM, Retrieval Augmented Generation, Dataset

## 1 INTRODUCTION

In modern mathematics, the increasing complexity of proofs has made peer review more difficult. Errors in proofs often go unnoticed for extended periods, as critical flaws are usually subtle and require expert scrutiny. As a solution, formal mathematical languages, also known as Interactive Theorem Provers (ITP), such as HOL Light (Harrison, 1996), Coq (Barras et al., 1999), Isabelle (Paulson, 1994), and Lean (Moura & Ullrich, 2021), allow for automated verification of proofs, reducing the risk of human oversight.

However, writing proofs in these formal languages requires significant effort and expertise. Mathematicians must navigate through unfamiliar theorem libraries and often engage in repetitive tasks due

---

[*]Equal contribution.

[†]Corresponding author.

[1]The model can be found at `https://huggingface.co/FrenzyMath/Herald_translator`. The code of Herald and LeanSearch can be found at `https://github.com/frenzymath/herald_translator`.

to the strict requirements of formal languages, which can be burdensome for those accustomed to writing high-level, natural language proofs.

This highlights the importance of autoformalization, which seeks to translate natural language (NL) reasoning into formal language (FL), with the reverse process referred to as autoinformalization, making 'natural' and 'informal' interchangeable in our text. Utilizing large language models (LLMs) is a promising approach to this task, as LLMs can learn reasoning patterns from large corpora of natural language mathematics, apply them to NL-FL translation, and add necessary reasoning steps in formal logic.

However, the scarcity of parallel data between natural and formal languages, which consists of one-to-one pairs aligning natural language with its formal language counterpart, limits the progress of LLM-based translation approaches. To address this scarcity, existing works explore methods such as using LLMs to annotate Lean corpora (Wang et al., 2024) and Expert Iteration (Ying et al., 2024a; Xin et al., 2024a). Yet, these methods do not fully leverage the detailed structural information provided by the Lean 4 compiler and the pyramid architecture of the Lean repository.

In this work, we introduce Herald (**Hi**erarchy and **R**etrieval-based Tr**a**nslated **L**ean **D**ataset), a dataset created by applying our augmentation pipeline to Mathlib4. During statement informalization, we provide the LLM with rich contextual information, especially theorem dependencies, and follow a hierarchical approach where dependent theorems are informalized before the target one, ensuring comprehensive natural language annotations. For proof informalization, we further enhance the LLM's understanding by offering term explanations for each translation step, supported by our NL-FL statement dataset. Additionally, we synthesize more formal statements by decomposing tactic-wise proofs into smaller steps, generating 580k valid statements from 110k original Mathlib4 theorems. We also utilize LLMs to generate NL counterparts for synthesized formal statements, further augmenting our NL-FL corpus.

Based on the Herald dataset, we fine-tuned a model for NL-FL statement translation. To validate the generated formalized statements, we apply both Lean compiler and LLM back-translation checks. Our model achieves 96.7% accuracy on miniF2F-test (Zheng et al., 2022) and 23.5% accuracy on our internal graduate-level textbook dataset, outperforming InternLM2-Math-Plus-7B (Ying et al., 2024b) (73.0% and 7.5%) and TheoremLlama (Wang et al., 2024) (55.0% and 4.0%). To demonstrate the model's effectiveness in autoformalization, we apply the Herald translator to a section of the Stack Project (See Appendix F.1), using DeepSeek-Prover-V1.5 (Xin et al., 2024b) to complete the proofs.

In conclusion, our contributions are as follows:

- We propose a structural-information-aware pipeline for augmenting NL-FL datasets from any Lean project. The inclusion of additional context and a hierarchical process breaks down formalization into manageable steps, improving LLM performance and enabling formalization at the project level, rather than focusing on individual theorems or files.

- We present the Herald dataset, generated from our pipeline on Mathlib4, containing 580k valid statements and 44k NL-FL theorem pairs.

- We release the Herald translator model, fine-tuned on the Herald dataset, achieving 96.7% accuracy on miniF2F-test and 23.5% on our internal graduate-level dataset, significantly outperforming InternLM2-Math-Plus-7B (73.0% and 7.5%) and TheoremLlama (55.0% and 4.0%).

## 2 RELATED WORK

**Auto-formalization**   The field of auto-formalization has advanced notably with the integration of LLMs. Early efforts, like Wang et al. (2018), trained specialized neural models for statement auto-formalization. Recent studies, including Wu et al. (2022), Patel et al. (2023), and Zhou et al. (2024), employ LLMs with few-shot in-context learning, while Agrawal et al. (2022) introduces input-dependent few-shot learning. Additionally, Azerbayev et al. (2023) and Jiang et al. (2023a) fine-tune LLMs on natural language-formal language pairs to improve accuracy without in-context learning.

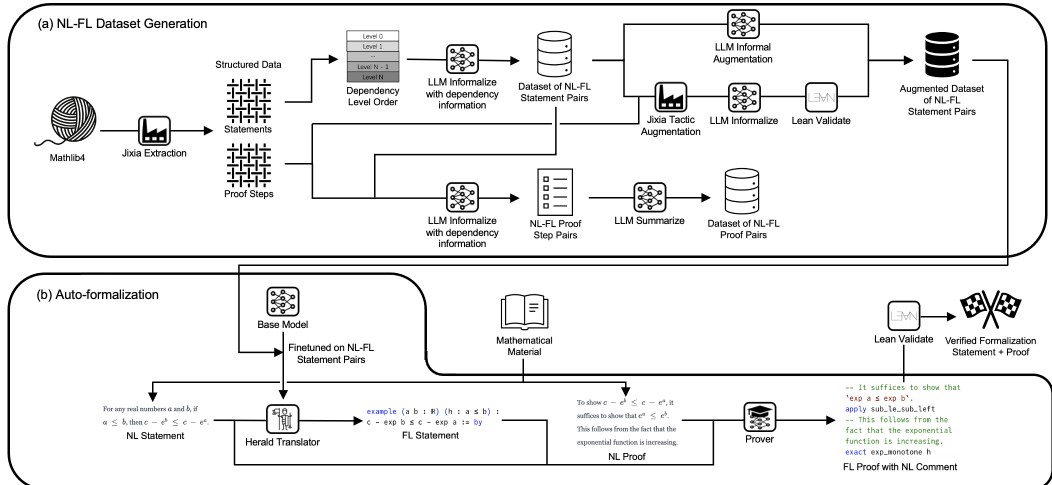

Figure 1: Overview of our approach. (a) NL-FL Dataset Generation: We extract statements from Mathlib4 and informalize them by providing the LLM with rich contextual information, particularly dependent theorems, and proceed in dependency-level order to ensure the LLM has access to all relevant natural language annotations. The same pipeline is applied to the proof corpus, aided by the NL-FL statement dataset generated in the previous step. Additionally, we augment the statement corpus in two ways: by breaking down tactic-wise proofs in Mathlib4, with results validated through the Lean compiler, and by using LLMs to generate equivalent NL statements. (b) Autoformalization Pipeline: We train a statement formalizer on the Herald dataset. During formalization, FL statements are first generated by the Herald translator and then fed into a powerful automatic theorem prover (e.g., DeepSeek Prover V1.5) to obtain the final formalized corpus.

Extending beyond statements, auto-formalization of proofs presents a more complex challenge. Jiang et al. (2023b) and Xin et al. (2024c) propose frameworks that use in-context learning to generate formal proof sketches from LLM-produced natural language proofs. These sketches are then complemented by auto-theorem-proving tools such as sledgehammer in Isabelle to fill in any gaps. Wang et al. (2024) and Shao et al. (2024) generate complete formal proofs with natural language in-line comments using fine-tuned LLMs, with Wang et al. (2024) also capable of translating both natural language statements and proofs.

**NL-FL dataset generation**    The pursuit of auto-formalization faces significant challenges primarily due to the shortage of high-quality and high-quantity NL-FL pairs as training data. Current efforts in generating these pairs still face substantial limitations.

Several approaches (Jiang et al., 2023a; Lin et al., 2024; Wang et al., 2024; Ying et al., 2024a) have recently attempted to address this issue by leveraging LLMs to generate NL-FL pairs. Specifically, MMA (Jiang et al., 2023a) uses LLMs to generate 88K NL statements, starting from formal statements extracted by the LeanDojo framework. Lean-STaR (Lin et al., 2024) takes a different approach by generating NL 'proof thoughts' at each step of a formal proof, producing 52,438 NL proof thoughts based on theorems in the LeanDojo library. TheoremLlama (Wang et al., 2024) enhances the process by introducing a bootstrapping technique where NL proofs are integrated into Lean4 code to create a training dataset. Lean Workbook (Ying et al., 2024a) proposes a novel pipeline that iteratively generates and filters synthetic data to translate natural language mathematical problems (extracted from math contest forums) into Lean 4 statements and vice versa.

Despite these efforts, the primary limitation of all the aforementioned methods lies in the intrinsic weaknesses of LLMs when applied to mathematics and logical reasoning. As a result, the generated NL-FL pairs are prone to errors, which can propagate through datasets and impair the performance of models trained on them. In this paper, we introduce a novel Retrieval Augmented Generation (RAG) pipeline specifically designed to ensure both the accuracy and naturalness of the natural language generated from formal mathematical statements and make our dataset more reliable for training auto-formalization models.

## 3 METHODOLOGY

In this section, we outline our methodology for constructing and utilizing the Herald dataset to improve LLMs' ability to translate mathematical statements and proofs between NL and FL. Our approach centers around generating high-quality NL-FL pairs from Mathlib4 through providing LLM with sufficient structural information, followed by strategic augmentation techniques to address data scarcity and distribution imbalance. Section 3.1 explains the generation process of NL-FL data, while Sections 3.2 and 3.3 describe our augmentation strategies and the training of our statement formalization model on the Herald dataset. These steps collectively contribute to enhancing the autoformalization performance of LLMs within the Lean4 environment.

### 3.1 NL-FL DATA GENERATION

This subsection details the process behind creating the Herald dataset, a large-scale collection of NL-FL language pairs specifically designed to enhance the performance of LLMs in autoformalization. Using Mathlib4 as our source of formal statements and proofs, we apply a RAG approach to produce high-quality natural language translations. The Herald dataset consists of 580k NL-FL statement pairs and 45k NL-FL proof pairs, making it one of the largest resources for training models on translating between natural and formal mathematical languages. This section describes the detailed extraction and augmentation methodologies that were employed to construct this dataset.

#### 3.1.1 STATEMENTS INFORMALIZATION

**Structured Information and Contextual Augmentation**    The first step in our methodology involves extracting essential components from Lean code that encapsulate formal statements. We utilize Lean-Jixia[2], a static analysis tool specifically designed for Lean 4, to extract structured information from Mathlib4. Lean-Jixia parses Lean files to extract key metadata, including theorem declarations, proof structures, and dependency relationships. We select five main components to enhance the FL to NL translation process: head statements, kind, docstrings, neighbor statements, and dependent theorems. These additional structures provide mathematical context and background information for the LLM, thereby reducing the burden on the LLM to infer the mathematical background independently. For an overview of each component, please refer to Appendix B. By utilizing this information, the LLM can better understand the FL statement and follow the principles of NL statements when translating FL statements into NL statements.

**Dependency Level and Translation Order**    We identify a critical issue where the lack of natural language translations for dependent definitions often led to incorrectly fabricated translations of these dependencies, thereby affecting the translation of the original theorem. To address this, we utilize Lean-Jixia to extract the dependency graph of all statements, forming a directed acyclic graph (DAG). We stratify all statements into levels based on their distance to the root nodes, which are statements not dependent on any others. Statements in each level only depend on those in lower levels. By translating statements in the level order, we ensure that the natural language translations of dependent theorems are available during the translation process, thereby providing missing dependent information that does not exist in the formal statements.

**Retrieving Related Instances**    Following previous work (Gao et al., 2024) on the semantic search engine of Mathlib4, we identified the theorem most similar to the one to be translated through 1,000 manually annotated examples. To elaborate, we represent the formal language of the manually annotated theorems as embeddings. The human-annotated theorem that is closest in distance to the theorem to be translated in this embedding space is then placed in the instruction set of the LLM, thereby enhancing the quality of translation.

By calculating the proximity between the embedding of the theorem that requires translation and the embeddings of the annotated examples, we can effectively determine the most relevant precedent or analogous theorem.

Incorporating the closest matching theorems into the LLM's instructions functions as a contextual anchor, which guides the model in understanding the specific mathematical domain and terminology

---

[2]https://github.com/reaslab/Lean-Jixia

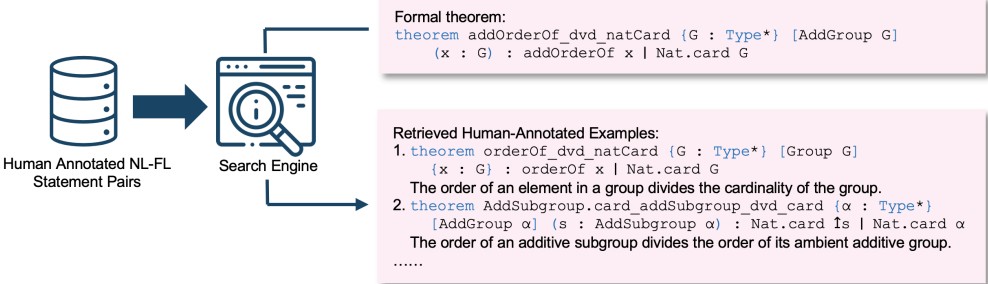

Figure 2: Illustration of how related instances are retrieved: NL-FL statement examples are embedded and stored in a vector database. The statement being informalized is treated as a query and embedded by the same model, which is designed to account for mathematical similarity. The vector database then retrieves a list of relevant theorems based on cosine similarity of the embeddings.

relevant to the new theorem. This context-aware guidance ensures that the LLM's translation maintains the technical precision and conceptual integrity of the original theorem.

**Human Feedback Iteration**  Human feedback is integral to refining and improving the translation process. We collect feedback from five human experts, all PhD students in pure mathematics with extensive expertise in Lean. These experts observe translation examples across various mathematical branches, identifying common issues in the natural language translations. They summarize these issues into principles and illustrative examples, which are then incorporated into the prompts used to guide the translation models. The iteration took place over six rounds of feedback and refinement, culminating in the development of over a dozen principles. By integrating these principles into the prompts, we ensure that the translations align with the precise and concise language habits of human mathematicians.

### 3.1.2 PROOFS INFORMALIZATION

In the previous section, we detailed the methodology for generating NL-FL statement pairs from Mathlib4. This section extends our approach to include the generation of NL-FL proof pairs, leveraging the same structure of principles and tools.

**Stepwise Translation and Integration**  The initial step in generating NL-FL proof pairs involves extracting proof line information using Lean-Jixia. Lean 4 supports two styles of proofs: tactic-based and term-based. A detailed comparison of these styles is provided in Appendix A. Term-based proofs, though concise, often present a series of formal theorems without a clear expression of logical reasoning. This makes them less effective in enhancing LLMs' inference capabilities during training, as they lack the logical chain that aids comprehension. Therefore, we focus on extracting and translating tactic-based proofs. We begin by translating each line of the formal proof into natural language. Once each line is translated, these individual translations are combined to form a complete informal proof.

This approach allows us to supplement the formal proof with extensive Lean information that would otherwise be missing, such as the proof state before and after each proof step. This additional context helps the LLM understand how each proof step contributes to the overall proof, thereby enhancing the model's comprehension and the accuracy of the NL translations.

**Structured Information for Proof Translation**  In addition to the structured information provided during the translation of statements, we further extract more detailed components for the translation of proofs and utilize the previous statement translation results. These additional components include: formal statement, informal statement, tactic information, and proof states. For an overview of each component, please refer to Appendix B.

**Tactic Explanation**    A significant limitation of LLMs in translating mathematical proofs is their lack of understanding of the logical relationships between the proof steps and the goal. To address this issue, we employ human annotation to explain the logical structure inherent in each type of tactic. By adding these detailed explanations of the logical structure into the prompts, we significantly enhance the logical coherence of the natural language translations of mathematical proofs.

## 3.2    AUGMENTATION

When training models on NL-FL pairs in the context of Lean 4, two major challenges arise: **Data Scarcity** and **Distribution Imbalance**. The limited availability of NL-FL pairs leads to model over fitting, while the deformalization process often produces informal statements that deviate from the natural distribution found in textbooks, resulting in rigid, repetitive, or overly precise content that hinders generalization.

To address these challenges, we introduce 2 innovative augmentation techniques designed to both expand the dataset, and align it more closely with the real-world distribution of mathematical statements, proving to be highly effective in practice.

### 3.2.1    TACTIC-BASED AUGMENTATION

Natural language theorem proving presents a significant challenge due to the inherent complexity of proofs, where theorems are often established through the interplay of various lemmas, techniques, and proof strategies. Learning the 'global' properties of theorems from such complex proofs is difficult because the entire structure is often too intricate for direct modeling. However, we observe that during the proof process, each proof step—often a tactic—addresses a smaller, localized 'statement', which is simpler and more easily understood in isolation. In fact, each such local statement is fully captured by the prove state (or tactic state) in Lean's interactive prover. Given this insight, we developed an

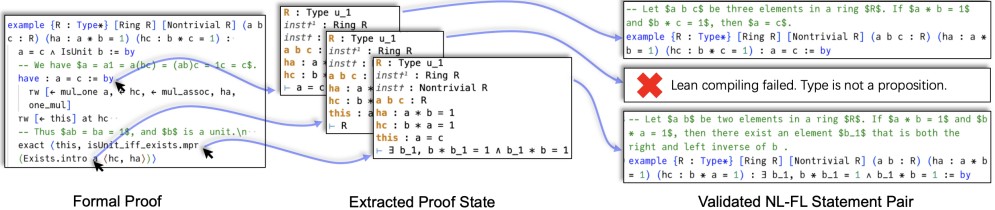

Formal Proof        Extracted Proof State        Validated NL-FL Statement Pair

Figure 3: Demonstration of Tactic Augmentation Strategy

augmentation strategy based on extracting prove states from Lean 4. Typically, the proof of theorem comprises of prove states or tactic states. For each prove state, which contains the conditions and the goal at that specific proof step, we construct a new formal language statement.

This augmentation strategy (Figure 3) ensures that each generated informal statement is aligned with a concrete, localized mathematical goal, making it both provable and semantically valid. In this way, from each proved theorem, we can generate multiple statements that is not only mathematically sound but also more straightforward and reflective of the structure found in real-world theorems.

**Deduplication** Nevertheless, in proofs involving many closely related tactics, consecutive tactic states can result in augmented statements that are highly similar to each other. Directly incorporating all of these similar statements into the training set risks overfitting the model. To address this, we randomly sample a subset of the augmented statements, equivalent in number to the original theorems, ensuring that the augmented dataset retains diversity and avoids excessive repetition of similar content.

### 3.2.2    AUGMENTATION VIA MATHEMATICS-PRETRAINED LLM

Our second method (Figure 4) capitalizes on LLMs pre-trained on extensive mathematical corpora. To ensure that the augmented natural language statements maintain both semantic consistency and variability, we employ LLM pre-trained on extensive mathematical data. This allows us to generate multiple equivalent informal statements for each formal statement.

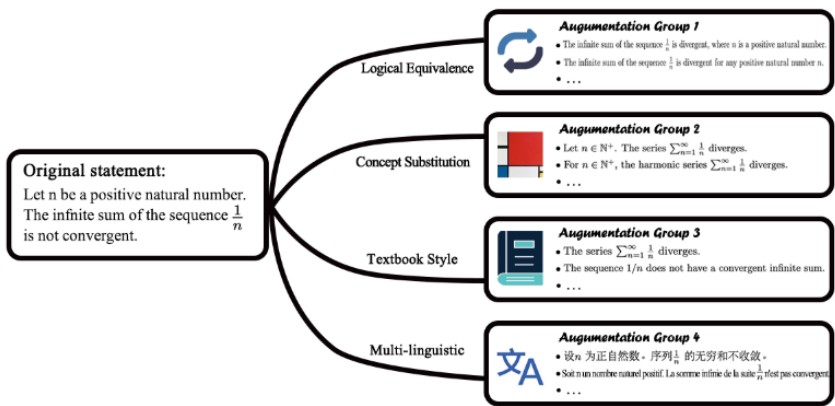

Figure 4: Demonstration of LLM Informal Augmentation

We implement four specific augmentation strategies:

- **Logical Equivalence Rewriting** For each formal statement, we generate several logically equivalent informal statements. For example, we might rewrite "If AA, then BB" as "BB holds given AA."

- **Abstract Concept Substitution** In some cases, lengthy or detailed informal statements can be rephrased using more abstract or higher-level mathematical concepts. For example, the statement 'For given matrix $A$, there exists a matrix $B$, such that $AB = BA = I$' would be replaced with a more concise, abstract expression or concept like '$A$ is non-degenerate'.

- **Omission of Implicit Condition** In natural language mathematical discourse, particularly in textbooks and research papers, certain conditions are often omitted because they are considered obvious or conventionally understood by the reader. For example, a theorem might not explicitly mention the requirement that a function be continuous if that is implied by the context.

- **Multi-linguistic Translation** We generate additional informal statements by translating the formal statements into Chinese, French and Russian. This results in a set of informal statements that represent how theorems might be articulated in a different language.

This approach benefits from the LLM's ability to capture and reproduce the natural distribution of mathematical language, as it has been pre-trained on a vast amount of mathematical text.

In this way, our approach can be viewed as a resampling of the original dataset. Furthermore, after generating the augmented statements, we again sample a number of statements equivalent to the original theorems, ensuring that the augmented dataset remains representative of the natural language distribution in mathematical texts and avoids over-representation of any specific phrasing or style.

### 3.3 TRAINING STATEMENT FORMALIZING MODEL ON THE HERALD DATASET

After establishing the Herald dataset (Table 1), we then perform Supervised Fine-Tuning (SFT) on a pre-trained LLM using this combined dataset. Training on NL-FL datasets can enhance the LLM's ability to translate natural language mathematical propositions into Lean4 propositions. Mixing in an appropriate proportion of general natural language data can prevent potential overfitting phenomena and catastrophic forgetting. This balanced approach ensures that the model maintains its general language understanding while developing specialized skills in formal mathematical translation.

| | Mathlib4 Original Statements | Augmented Statements | Mathlib4 Proofs |
|---|---|---|---|
| Number of NL-FL pairs | $291k$ | $580k$ | $44k$ |

Table 1: Statistics of Herald dataset and Mathlib4

## 4 EXPERIMENTS

We conduct extensive experiments to evaluate the Herald dataset and translator. In Section 4.1, we test the Herald translator on three statement datasets from different topics and compare its performance with other formalization models. Section 4.2 assesses the quality of the Herald dataset through expert inspection, while in Section 4.3, we apply our autoformalization pipeline to a section of the Stack Project and analyze the results. Ablation study on the two augmentation methods is presented in Appendix C.

### 4.1 STATEMENT FORMALIZING MODEL

#### 4.1.1 DATASET AND TRAINING

We selected DeepSeek-Prover-V1.5-Base 7B as our base model due to its extensive training on formal programming languages like Lean, which provides a strong foundation for formal reasoning tasks.

Our data preparation process involved several key steps to ensure a comprehensive and balanced dataset. We began by collecting 580k NL-FL pairs from the Herald dataset. From this, we created two datasets: one for translating informal to formal (NL→FL) mathematical statements and another for the reverse direction (FL→NL). This process yielded a total of 1.16M examples. The distribution of examples followed a 1:2:1 ratio among original statements, tactic-augmented data, and informal-augmented data. To further enhance model performance and mitigate overfitting or catastrophic forgetting, we combined our NL→FL and FL→NL datasets with the OpenHermes2.5 dataset (Teknium, 2023), a general-domain dataset. The final training data maintained a 2:2:1 ratio among NL→FL, FL→NL, and OpenHermes2.5 examples, respectively, for fine-tuning.

Our fine-tuning process consisted of two stages: first, we conducted a 2000-step warm-up using the OpenHermes2.5 dataset, followed by training on the mixed dataset. We used a learning rate of 4e-5 with a cosine decay schedule across 5 training epochs.

#### 4.1.2 VALIDATION PIPELINE

For validation, we adopt the pipeline from the Lean Workbook (Ying et al., 2024a), which includes several key steps:

1. **Translation**: Using our trained model to translate informal statements from the test set into formal statements.
2. **Validation**: Using a REPL (Read-Eval-Print Loop) based framework to verify that the translated Lean 4 statements are valid and pass compiler checks. This step ensures that our translations are syntactically correct in Lean4.
3. **Back-translation**: For statements that pass the validation in step 2, we used InternLM2-Math-Plus-7B to translate the formal statements back to natural language to assess the preservation of meaning.
4. **NLI check**: We use the DeepSeek Chat v2.5 model to compare the back-translated statements with the original informal statements, ensuring that our translations are mathematically accurate and preserve the intended meaning.

For a detailed discussion and case study of the validation pipeline's effects, especially the NLI check, please refer to Appendix D. We conducted human evaluations on the auto-formalization results of ProofNet (Azerbayev et al., 2022). The results indicate that while the pipeline is not perfectly accurate for evaluating informal-to-formal alignment, it remains a common practice in the field and one of the few practical approaches for large-scale automated evaluation at this time.

We perform 128 parallel translations and consider the translation successful if any of these passes both the compiler check and the nil check. Our results will be shown in the next subsection.

#### 4.1.3 RESULT

To evaluate the performance of our model, we conducted comprehensive tests comparing Herald with several models in similar settings. Our test suite included a diverse range of datasets:

| Model | miniF2F | | Extract Theorem | College CoT |
| --- | --- | --- | --- | --- |
| | test | valid | | |
| TheoremLlama | 50.1% | 55.6% | 4.0% | 2.9% |
| InternLM2-Math-Plus-7B | 73.0% | 80.1% | 7.5% | 6.5% |
| Llama3-instruct | 28.2% | 31.6% | 3.6% | 1.8% |
| **Herald** | **96.7%** | **96.3%** | **23.5%** | **16.0%** |

Table 2: Performance comparison of different models across various datasets. The last two datasets (Extract Theorem and College CoT) are shuffled subsets of 200 samples each.

**miniF2F (Zheng et al., 2022)** A widely-used benchmark dataset for formal mathematics.

**Extract Theorem** A custom dataset compiled by extracting theorems from advanced undergraduate-level textbooks using OCR on scanned materials. It covers a wide range of mathematical topics and includes multilingual content.

**College CoT** A curated dataset derived from digital mathematics resources across the internet, with content verified and filtered using a large language model (LLM) to ensure quality and relevance.

These datasets were carefully chosen to assess the models' capabilities across various levels of difficulty and categories of mathematics. For representative examples in the Extract Theorem and College CoT dataset, please refer to Appendix G.

**Note:** For models lacking header generation capability, we manually added a generic header. In cases where models couldn't output stable Lean statements, we truncated their generation to obtain a maximal possible statement (derived from generated proof or Lean code) for testing purposes.

We present the main experimental results in Table 2. From the table, we observe that the Herald translator achieves a success rate of 96.3% on MiniF2F-Valid, 96.7% on MiniF2F-Test, 23.5% on Extract Theorem, and 16.0% on College COT, demonstrating significant improvement on the MiniF2F dataset and a breakthrough on two new datasets, outperforming all baselines. These results indicate that the Herald translator excels at formalizing natural language statements, particularly in challenging tasks and high-level mathematics. A case study of the Herald translator's auto-formalization results is presented in Appendix D.

## 4.2 SUMMARY OF CASE STUDY IN HERALD QUALITY EXAMINATION

To assess the mathematical rigor and language style in the Herald dataset, we conducted several case studies. For a comprehensive result, see Appendix E.

In summary, the informal data in Herald demonstrates significant advantages in mathematical rigor and alignment with formal proofs. Lean-jixia extracts more complete Lean information (e.g., theorem names, variables), which aids in generating more precise theorem descriptions.

For statements, the relevant definitions are expanded and explained in natural language, enhancing the connection between natural language mathematics and Mathlib4. The language style is natural, with the logic of the statement properly expressed in the translation. However, there are instances where notations and formulas are either well-written in commonly used mathematical forms or poorly copied from formal language. Translating more abstract and diagram-based theorems remains a challenge.

The accuracy of statement translation contributes to the accuracy of proof translation. By starting with a stepwise translation, we ensure that the LLM-generated proof faithfully reflects the proof strategy used in the formal proof. Unlike a verbatim translation of each tactic's function, the proof connects these steps with logical reasoning. Formal theorems used in the proof are also explained. The level of detail in the translated proofs is closer to formal language rather than natural language. Similar to statements, some notations are well-written, while others are copied from formal language.

### 4.3 AUTOFORMALIZATION PRACTICE

To evaluate the performance of our Herald translator in real-world formalization tasks, we applied the proposed autoformalization pipeline (Figure 2.(b)) to a section of the Stacks Project, using **DeepSeek Prover 1.5** (Xin et al., 2024b) as the prover. The Stacks Project (Stacks Project Authors, 2018) is an open-source, collaborative online encyclopedia focused on modern algebraic geometry and related fields, making it an ideal test case for complex formalization challenges.

To demonstrate the capabilities of the Herald translator in auto-formalizing modern mathematics, we selected the Normal Extensions section (Stacks Project Authors, 2018, Tag 09HL) from the Field Theory chapter of the Stacks Project. This section was successfully formalized into a runnable Lean 4 source file, showcasing the effectiveness of our pipeline. For more details on the Stacks Project, see Appendix F.1, and for the formalized output, refer to Appendix F.2.

The generation was completed efficiently using a 16-pass setting, with human checks revealing only two necessary theorem modifications: correcting the conclusion in two of the auto-formalized theorems, and removing an unnecessary condition in one of them. Notably, the model demonstrated strong understanding of the content, achieving both mathematical and programming correctness. For prover configuration, we use DeepSeek-Prover-V1.5-RL + RMaxTS with $4 \times 512$ sample budget), which successfully proved only one two-line theorem relying on an existing lemma from Mathlib4. This highlights the need for a more capable prover model to handle advanced topics, a key focus of our future work.

## 5 CONCLUSIONS

In this paper, we present Herald, a structural-information-aware pipeline for generating a rich NL-FL dataset from Lean projects, specifically Mathlib4. Our approach augments the traditional process by providing hierarchical, context-rich annotations, ensuring that dependencies are fully accounted for before translating target theorems. This methodology not only facilitates better natural language explanations but also breaks down complex formal proofs into more manageable components, improving the performance of LLMs in the formalization process.

We release the Herald dataset, which includes 580k valid statements and 44k NL-FL theorem pairs, providing a significant resource for formalization research. Additionally, we fine-tuned a statement formalizer on this dataset, which achieves state-of-the-art accuracy— 96.7% on the miniF2F-test and 23.5% on a graduate-level textbook dataset—substantially outperforming existing baselines such as InternLM2-Math-Plus-7B and TheoremLlama. By applying our translator to a section of the Stack Project and leveraging the DeepSeek Prover V1.5, we further demonstrate the practical viability of our approach in auto-formalization.

Our work contributes to the field in several ways: the development of a scalable NL-FL dataset generation process that incorporates hierarchical dependencies, the introduction of the Herald dataset, and the release of a high-performing translation model. These contributions mark a significant step toward automating formalization tasks at a project-wide scale, and we believe the methodologies and resources presented here will facilitate further advancements in the field of mathematical formalization and LLM-based theorem proving.

## 6 ACKNOWLEDGEMENTS

This work is supported in part by the New Cornerstone Investigator Program and in part by Ubiquant.

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

## A    INTRODUCTION TO FORMAL LANGUAGES

There are various approaches and tools of formalized mathematics. Among them, type-theory based theorem provers are prominent. They include Lean (Moura & Ullrich, 2021), Coq (Barras et al., 1999), Agda (Norell, 2009), Isabelle (Paulson, 1994), and many others.

Type-theory based provers utilize Curry–Howard Isomorphism, or the Proposition-as-Type paradigm (Wadler, 2015) to encode mathematical statements as types. In this paradigm, proofs for a statement $P$ are no different from a programmatic value of type $P$. The practice of writing mathematical proofs is thus unified with the practice of programming.

Interactive theorem provers also have a distinct tactic mode, where the prover keeps track of currently available hypotheses and the goal to be proved. A set of concise, mid- to high-level commands called *tactics* can be used to manipulate the state. When all goals are solved, the proof is considered to be complete and the prover automatically generates the low-level code. The tactics are often designed to reflect common patterns in natural language proofs, making it easier for the users to write and understand proofs.

Thus, in tactic mode, the user simply writes a sequence of tactics to describe the steps (i.e., the tactic-based proof) required to prove the problem at hand, rather than manually writing down the detailed term-based proofs.

Lean 4 (Moura & Ullrich, 2021) is an interactive, type-theory based prover. Lean 4 is designed to be highly extensible via its metaprogramming capability, enabling Lean-Jixia to extract important metadata accurately and concisely.

Mathlib4 (mathlib Community, 2020) is a community-driven effort to build a unified library of mathematics in Lean 4. It has a substantial part of modern mathematics formalized. Hence it can act as a reliable source for LLMs to learn research-level mathematics.

Below is a comparison of the two styles:

```
example (p q r : Prop) (h : p ∧ q ∧
    r) : q ∧ p ∧ r :=
  And.intro h.right.left
    (And.intro h.left
    h.right.right)
```

Figure 5: Term-based Proof

```
example (p q r : Prop) (h : p ∧ q ∧
    r) : q ∧ p ∧ r := by
  rcases h with ⟨hp, hq, hr⟩
  constructor
  · exact hq
  · constructor
    · exact hp
    · exact hr
```

Figure 6: Tactic-based Proof

Tactic-based proofs are usually more informative and closer to natural language proofs. Besides, one has access to proof states in tactic mode, which contains valuable information to help the informalization process.

## B    STRUCTURED INFORMATION IN LEAN FOR RAG

Here, we provide a detailed explanation of each component of the structured information we provide to the LLM, which is used to enhance the FL to NL translation process in data generation. During proof translation, all structured information (excluding kind) for statements is provided, along with additional components to further assist the LLM.

COMPONENTS FOR STATEMENT TRANSLATION

- **Head statements**. These include foundational theorems, definitions, and other significant statements within the mathematical field relevant to the theorem. The extraction of head statements ensures that the context and background of the theorem are well-understood by the LLM.

- **Kind.** The kind of statement, which can be a `theorem`, `instance`, `definition`, `structure`, `class`, `inductive`, `classInductive`, or `opaque`, provides essential information about the nature of the statement. This classification aids the LLM in applying appropriate translation strategies tailored to the specific type of mathematical entity being processed. Different prompts are employed for translating different kinds of statements, adapting to the varying language habits of mathematical statements.

- **Docstrings** often contain NL explanations written by humans, which are crucial for translating formal statements into more understandable language. The LLM is trained to leverage these docstrings effectively, ensuring that all relevant mathematical information is included while filtering out implementation notes that are not pertinent to the translation.

- **Neighbor statements** in the Lean code, including those with similar names or located within the same namespace or file, are indicative of related theorems or definitions. By considering these neighbor statements, the LLM can better understand the interconnectedness of mathematical concepts and ensure that the translation reflects these relationships.

- **Dependent theorems.** The inclusion of dependent theorems is crucial for maintaining the logical integrity of the translation. These theorems form the basis upon which the main theorem is built, and their inclusion ensures that the LLM can accurately reflect the proof's logical flow and dependencies.

ADDITIONAL COMPONENTS FOR PROOF TRANSLATION

- **Formal Statement**. The formal statement being proved.

- **Informal Statement**. The informal translation of the formal statement generated by LLM provides a natural language overview. This natural language translation is precisely the result generated as described in the previous section.

- **Tactic Information**. Details about the tactics used, including their effect in proving the theorem and how they should be translated into natural language. This is provided for each proof step.

- **Proof States**. Intermediate proof states presenting the current goal after using a tactic and how values of variables change by using the tactic, ensuring a comprehensive view of the proof's progression.

## C  ABLATION STUDY

In this section, we conduct an ablation study on the two augmentation methods. We trained 5 models on different composition of datasets. These training sets share the same "original" set which denotes the informal-formal pairs chosen from the translated Mathlib4 dataset, and we ablate the augmented datasets by adjusting different ratios between the "original" set and augmented statements to assess the effectiveness of our methods.

| Dataset ratio | Dataset size | MiniF2F-test | ProofNet-valid | Extract-Theorem |
|---------------|--------------|--------------|----------------|-----------------|
| 1:0:0 | 145k | 86.8% | 50.0% | 15.0% |
| 1:1:1 | 435k | 91.4% | 68.3% | 17.5% |
| 1:2:0 | 435k | 95.1% | 74.7% | 19.0% |
| 1:2:1 | 580k | 93.2% | 78.6% | 20.5% |
| 1:2:1 | 1160k | 96.7% | 81.6% | 23.5% |

Table 3: Performance comparison of Herald translator trained on different size and composition of datasets across various datasets. Dataset ratio represents 'original : tac_aug : inf_aug', **original** denotes the informal-formal pairs chosen from the translated Mathlib4 dataset, **tac_aug** refers to the tactic-based augmented statements, and **inf_aug** corresponds to the informal augmented statements. The penultimate model is trained on NL→FL dataset, and the last model, as our model in the main experiment, is trained on NL→FL and FL→NL datasets.

As shown in table 3, our augmentation methods effectively enhance model performance, particularly on more challenging datasets such as ProofNet. The model trained solely on the original informal-formal pairs (1:0:0) achieves a respectable accuracy of 50.0% on the ProofNet-valid dataset, 86.8% on the MiniF2F-test dataset, and 15.0% on the Extract-Theorem dataset, which underscores the quality of the informalized Mathlib4 dataset.

The introduction of augmented datasets significantly boosts model performance across all datasets. The model trained with a 1:1:1 ratio shows a notable improvement, achieving 68.3% on ProofNet-valid, 91.4% on MiniF2F-test, and 17.5% on Extract-Theorem. The 1:2:0 ratio model, which includes more tactic-based augmented statements, further enhances performance, reaching 74.7% on ProofNet-valid, 95.1% on MiniF2F-test, and 19.0% on Extract-Theorem. The model trained with 1:2:1 ratio on both NL→FL and FL→NL datasets outperforms the model trained with 1:2:1 ratio on only NL→FL datasets, with accuracies of 81.6% on ProofNet-valid, 96.7% on MiniF2F-test, and 23.5% on Extract-Theorem. This result indicates that increasing the size of the training set, while maintaining a balanced augmentation ratio, leads to substantial improvements in model performance, especially on more challenging datasets.

## D  DISCUSSION AND CASE STUDY FOR VALIDATION PIPELINE

In this section, we discuss the soundness and fairness of our validation pipeline and analyze examples of passing and failing the validation pipeline.

### DISCUSSION

It is important to note that even the strictest LLM-based NLI checking introduces false positives. The NLI check consists of two steps: back translation and semantic comparison. During back translation, LLMs are less capable than humans at detecting subtleties in formalized statements, such as filling back opened namespaces and disambiguating overloading notations. This may result in errors in back translation. During semantic comparison, the LLM may fail to decide correctly due to a lack of mathematical understanding and the various possibilities of stating mathematics in neutral language. Consequently, formalized results passing the NLI check may fail to reflect the mathematical meaning of the original natural language statement (i.e., false positives). Therefore, NLI checking is not as reliable as human expert evaluation.

However, while NLI checking is not perfectly accurate for evaluating informal-to-formal alignment, it remains a common practice in the field (e.g., see Ying et al. (2024a)) and one of the few practical approaches for large-scale automated evaluation at this time. It is not practical to apply human evaluation to a dataset with more than 1k formalization results. It is also worth noting that NLI check may introduce false negatives, i.e., results rejected by the NLI check that are actually correct.

For our application of the NLI checking pipeline, the most important goal is to create a fair performance test between different models. As we will see in the following experiments, both the baseline model and our model achieve similar results in terms of the proportion of false-positive cases.

To provide a more illustrative result on false positive cases and fairness mentioned in the above discussion, we conducted an experiment to evaluate the results passing the validation pipeline.

### EXPERIMENT RESULTS

We choose the ProofNet dataset to conduct our experiment. The ProofNet dataset consists of problems drawn from popular undergraduate mathematics textbooks, selected based on the following criteria: self-containment, ease of formalization, and low risk of train-test overlap.

We perform 128 parallel translations of the ProofNet validation set (with 185 natural language statements) using the InternLM2-Math-Plus-7B and Herald model. The translation is considered as validation accepted if any of these passes both the compiler check and the nil check and we only collect the first passed formalization for human evaluation.

The validation-passed auto-formalization outcomes are categorized by human experts into three distinct groups: correct translations, minor errors, and major errors. A correct translation must accurately reflect the mathematical meaning of the natural language statement. In cases where the original statement is ambiguous, we allow the formalized statement to adopt any mathematically valid interpretation. If the formalized result deviates from the original statement but can be easily

corrected by altering a single formal definition or restructuring a single logical relation within the hypothesis or statement, it is classified as a minor error. All other cases are classified as major errors. In section D.1, we will present examples of these three categories.

| Model | Validation Passed | Correct Translation | Minor Error | Major Error |
|---|---|---|---|---|
| InternLM2-Math-Plus-7B | 72 | 42 | 12 | 18 |
| Herald | 151 | 101 | 24 | 26 |

Table 4: Human evaluation results of validation passed statements.

As shown in Table 4, other baseline models also suffer from false positives, with comparable false positive rates, making the comparison fair.

For a more thorough inspection, in subsection D.2, we will present examples that are rejected by the Lean validation step and the NLI validation step. For false negative cases, given the substantial number of translation attempts, it is impractical to manually detect all such cases. Therefore, we will only provide one example in subsection D.2.3.

During the evaluation process, we also identified some potential mathematical inaccuracies or inaccurately truncated problems within the ProofNet dataset, which are detailed in Appendix D.3.

### D.1 RESULTS ACCEPTED BY THE VALIDATION PIPELINE

#### D.1.1 CORRECT TRANSLATION

Rudin|exercise_3_20

INFORMAL STATEMENT

Suppose $\{p_n\}$ is a Cauchy sequence in a metric space $X$, and some sequence $\{p_{nl}\}$ converges to a point $p \in X$. Prove that the full sequence $\{p_n\}$ converges to $p$.

GENERATED FORMAL STATEMENT

```
import Mathlib
open Filter
open Topology Filter

theorem full_convergence_of_subsequence {X : Type*} [MetricSpace X] {p
    : X}
  {s : ℕ → X} (hs : CauchySeq s) (l : ℕ → ℕ) (hl : Tendsto l atTop
    atTop)
  (hpl : Tendsto (s ∘ l) atTop 𝒩( p)) : Tendsto s atTop 𝒩( p) := sorry
```

ANALYSIS

This is an example where the Herald translator correctly translated the subtle condition "some sequence $\{p_{nl}\}$" in natural language. To be a subsequence, the index $nl$ must tend to infinity, which is reflected in the condition (hl : Tendsto l atTop atTop).

Axler|exercise_5_12

INFORMAL STATEMENT

Suppose $T \in \mathcal{L}(V)$ is such that every vector in $V$ is an eigenvector of $T$. Prove that $T$ is a scalar multiple of the identity operator.

GENERATED FORMAL STATEMENT

```
import Mathlib
open LinearMap
open Submodule

--We are to prove that if every vector is an eigenvector,then T is a
    scalar multiple of identity

theorem isEigenvector_spec (V : Type*) [AddCommGroup V] [Field K]
    [Module K V]
    (T : V →ₗ[K] V) (h : ∀ v : V, ∃ k : K, k · v = T v) : ∃ k : K, ∀ v
    : V, k · v = T v := by sorry
```

ANALYSIS

In this example, the formalized condition (h : ∀ v : V, ∃ k : K, k • v = T v)
for "every vector in $V$ is an eigenvector" and the formalized result
∃ k : K, ∀ v : V, k • v = T v for "$T$ is a scalar multiple of the identity opera-
tor" differ only in the order of quantifiers. The Herald translator handled this difference in the order
of quantifiers appropriately.

### D.1.2  MINOR ERROR

`Ireland-Rosen|exercise_18_4`

INFORMAL STATEMENT

> Show that 1729 is the smallest positive integer expressible as the sum of two different integral
> cubes in two ways.

GENERATED FORMAL STATEMENT

```
import Mathlib
open Finset

theorem smallest_two_way_sum_of_cubes_is_1729 :
    IsLeast {n : ℕ | ∃ a b c d : ℤ, a ≠ c ∧ a^3 + b^3 = c^3 + d^3 ∧ n =
    a^3 + b^3} 1729 := sorry
```

ANALYSIS

In natural language, we treat $1729 = a^3 + b^3$ and $1729 = b^3 + a^3$ as the same cube decomposition.
Therefore, the condition a ≠ c is not sufficient to guarantee two distinct cube decompositions.

`Rudin|exercise_1_19`

INFORMAL STATEMENT

> Suppose $a, b \in R^k$. Find $c \in R^k$ and $r > 0$ such that $|x-a| = 2|x-b|$ if and only if $|x-c| = r$.
> Prove that $3c = 4b - a$ and $3r = 2|b - a|$.

GENERATED FORMAL STATEMENT

```
import Mathlib
open Topology Filter Set

theorem problem_1_19 (a b : ℝ) : ∃ c r : ℝ, |x - a| = 2 * |x - b| ↔ |x
    - c| = r ∧ 3 * c = 4 * b - a ∧ 3 * r = 2 * |b - a|   :=  by sorry
```

ANALYSIS

There is a logical flaw in the formalized statement. The formalized statement is equivalent to saying that if $3c = 4b - a$ and $3r = 2|b - a|$, then $r$ and $c$ satisfy the previous conditions. However, the converse direction, namely that any $r$ and $c$ satisfying the previous conditions must also satisfy $3c = 4b - a$ and $3r = 2|b - a|$, is not translated.

### D.1.3 MAJOR ERROR

`Dummit-Foote|exercise_9_4_11`

INFORMAL STATEMENT

> Prove that $x^2 + y^2 - 1$ is irreducible in $\mathbb{Q}[x, y]$.

GENERATED FORMAL STATEMENT

```
import Mathlib
open Ideal Ideal.Quotient Finset

theorem irreducible_X_pow_two_add_Y_pow_two_sub_one : Irreducible (X ^
    2 + Y ^ 2 - 1 : Polynomial ℚ) := sorry
```

ANALYSIS

This is a major error that is difficult to detect. Lean can automatically deduce the type of variables `X Y` as $\mathbb{Q}$-coefficient polynomials, allowing this statement to pass the Lean check. However, we intended `X Y` to be the special generators in the ring of polynomials (also in the ring of polynomials in two variables instead of one). This is indeed an incorrect formal statement.

`Dummit-Foote|exercise_3_4_11`

INFORMAL STATEMENT

> Prove that if $H$ is a nontrivial normal subgroup of the solvable group $G$ then there is a nontrivial subgroup $A$ of $H$ with $A \trianglelefteq G$ and $A$ abelian.

GENERATED FORMAL STATEMENT

```
import Mathlib

section exercise_3_4_11

--Prove that if $H$ is a nontrivial normal subgroup of the solvable
    group $G$ then there is a nontrivial subgroup $A$ of $H$ with $A
    \unlhd G$ and $A$ abelian.

--We will use the fact that a group of order $p_1^{e_1} \cdots
    p_k^{e_k}$ has a nontrivial subgroup of order $p_i^{e_i}$, where $
    p_i$ are primes and $e_i > 0$.

--We proceed by induction on the derived series of $G$.

--If $G$ is abelian, then $G$ itself is a nontrivial normal subgroup of
    itself, and we can take $A = G$.

--If $G$ is nonabelian, then $G$ has a nontrivial normal subgroup $H$
    of its center $Z(G)$, by lemma 1.

--By induction hypothesis, $H$ has a nontrivial abelian subgroup $A$ of
    it.

--Since $A \leq H$ and $H \leq Z(G)$, we have $A \leq Z(G)$.
```

```
--Since $A \leq Z(G)$ and $A \unlhd G$, we have $A \unlhd G$.

--Thus, we have found a nontrivial abelian subgroup $A$ of $G$ with $A
    \unlhd G$.

--Now, we need to show that $A$ is nontrivial.

--If $A$ is trivial, then $A$ is a cyclic group of order $0$.

--However, this contradicts the fact that $A$ is a subgroup of $H$,
    which is a nontrivial group.

--Thus, $A$ must be nontrivial.
end exercise_3_4_11
```

### ANALYSIS

We present this extremely unusual bad output here. This is the only output that did not write a single line of Lean code with mathematical meaning. All the outputs are written in Lean comments (thus containing no errors and passing the Lean check). The comments are indeed related to the original problem, thereby circumventing the NLI check.

### D.2 RESULTS REJECTED BY THE VALIDATION PIPELINE

The following examples are collected from all 128 formalization attempts of a statement. Therefore, there are multiple outputs corresponding to one single informal statement.

#### D.2.1 EXAMPLES REJECTED BY LEAN CHECK

`Munkres|exercise_32_1`

##### INFORMAL STATEMENT

Show that a closed subspace of a normal space is normal.

##### GENERATED FORMAL STATEMENT 1

```
import Mathlib
open Set Filter Function Topology List

theorem isNormal_subspace_of_isClosed {X : Type*} [TopologicalSpace X]
    [NormalSpace X]
    {S : Set X} (hS : IsClosed S) : @NormalSpace S := by sorry
```

##### ANALYSIS

The formal statement uses `@NormalSpace` but does not provide sufficient explicit arguments.

##### GENERATED FORMAL STATEMENT 2

```
import Mathlib
open Set Filter Function Topology

theorem isNormal_subspace_of_normalSpace {t : TopologicalSpace X}
    [NormalSpace X] (S : Set X)
    (hS : IsClosed S) : IsNormal S := sorry
```

##### ANALYSIS

The predicate `IsNormal` does not exist in Mathlib. This is a model-created predicate.

### D.2.2 EXAMPLES REJECTED BY NLI CHECK

`Munkres|exercise_32_1`

INFORMAL STATEMENT

> Show that a closed subspace of a normal space is normal.

GENERATED FORMAL STATEMENT 1

```
import Mathlib
open Set Filter Function Topology List

theorem munkres_32_1 {X : Type*} [TopologicalSpace X] {S : Set X}
    [NormalSpace X]
    (T : Set S) (hT : IsClosed T) : NormalSpace S := sorry
```

BACK TRANSLATION 1

> Let $X$ be a topological space; let $S$ be a subspace of $X$. Suppose that $X$ is normal. Show that $S$ is normal.

NLI OUTPUT 1

> Upon examining the two problems, we find the following differences:
>
> 1. **Scope of the Subspace**:
>
> - **Problem 1**: Specifies that the subspace must be "closed."
>
> - **Problem 2**: Does not specify any conditions on the subspace $S$ other than it being a subspace of $X$.
>
> These differences indicate that the problems are asking for different proofs or considerations due to the condition of the subspace in each problem.
>
> ——different——

ANALYSIS

The NLI check identifies that the condition $S$ is closed is missing, thereby rejecting the translation. Note that during the back translation, the unused variable $T$ is automatically dropped.

GENERATED FORMAL STATEMENT 2

```
mport Mathlib
open Set Filter Function Topology List

theorem munkres_32_1 {Y : Type*} [TopologicalSpace X] [TopologicalSpace
    Y]
    [NormalSpace X] (f : Y → X) (hf : Continuous f) : NormalSpace Y :=
    sorry
```

BACK TRANSLATION 2

> Let $X$ and $Y$ be topological spaces; let $S$ be a closed subset of $X$. Suppose that $X$ is normal (or $T_4$), and that $Y$ is normal (or $T_4$). Let $f$ be a continuous map of $S$ into $Y$. Prove that $S$ is normal (or $T_4$).",

NLI OUTPUT 2

> Let's analyze the statements in each problem:
>
> **Problem 1:**
>
> - Statement 1: A closed subspace of a normal space is normal.
>
> **Problem 2:**
>
> - Statement 1: Let $X$ be a normal space.
>
> - Statement 2: Let $Y$ be a metric space.
>
> - Statement 3: If $f : Y \to X$ is a continuous mapping, then $Y$ is a normal space.
>
> Comparing the statements:
>
> - Problem 1 focuses on the property of a closed subspace within a normal space, asserting that such a subspace is also normal.
>
> - Problem 2 introduces a normal space $X$ and a metric space $Y$, and discusses the implications of a continuous mapping from $Y$ to $X$ on the normality of $Y$.
>
> The statements in the two problems are different because Problem 1 deals with the normality of subspaces, while Problem 2 deals with the effect of a continuous mapping on the normality of a space.
>
> ——different——

ANALYSIS

The NLI check identifies the wrongly added information about $Y$ and $f$ and rejects the formal statement. However, note that the back translation is correct, but the analysis of statement 2 in the NLI output is not accurate. The condition that $Y$ is a metric space is never mentioned in the back translation.

### D.2.3   A FALSE NEGATIVE EXAMPLE

`Munkres|exercise_32_2b`

INFORMAL STATEMENT

> Show that if $\prod X_\alpha$ is regular, then so is $X_\alpha$. Assume that each $X_\alpha$ is nonempty.

GENERATED FORMAL STATEMENT

```
import Mathlib
open Set Function Filter TopologicalSpace
open scoped Topology

theorem regularSpace_of_product_regularSpace {ι : Type*} {X : ι →
    Type*} [∀ i, TopologicalSpace (X i)]
    [∀ i, Nonempty (X i)] (h : RegularSpace (Π i, X i)) : ∀ i,
    RegularSpace (X i) := sorry
```

BACK TRANSLATION

> Let $X_i$ be a regular space for each $i \in I$. Prove that the product space $\prod_{i \in I} X_i$ is regular.

In this example, the Herald translator accurately formalized the statement. The logic in the informal statement and the generated formal statement is the same: the product space being regular implies each component being regular. However, the back translation is incorrect, reversing the order of the implication. This results in a failed NLI check.

### D.3 POSSIBLE INACCURACIES IN THE PROOFNET DATASET

Here, we present some of the potential inaccuracies identified in the ProofNet validation set during human expert evaluation. We only focus on inaccuracies that affect the meaning or validity of the statements, and we do not include typographical errors in this list.

`Axler|exercise_6_16`

> Suppose $U$ is a subspace of $V$. Prove that $U^\perp = \{0\}$ if and only if $U = V$

In general, this statement does not hold. To make it valid, it suffices to add one of the following conditions:
(1) $V$ is finite-dimensional, or
(2) $U$ is finite-dimensional, or
(3) $U$ is a closed subspace of $V$.

`Putnam|exercise_1998_a3`

> Let $f$ be a real function on the real line with continuous third derivative. Prove that there exists a point $a$ such that

This statement is truncated at an inappropriate point.

`Rudin|exercise_4_8b`

> Let $E$ be a bounded set in $R^1$. Prove that there exists a real function $f$ such that $f$ is uniformly continuous and is not bounded on $E$.

This statement is mathematically wrong. A uniformly continuous function defined on a bounded subset is always bounded.

`Axler|exercise_1_3`

> Prove that $-(-v) = v$ for every $v \in V$.

This statement lacks a proper condition on $V$, making it too ambiguous. Without specifying the nature of $V$, one could assume it to be an additive group, vector space, or any other algebraic structure.

## E  CASE STUDY OF HERALD DATASET

In the following sections, we will first analyze examples of original statements, followed by the proofs, and conclude with the augmented statements. In sections E.1 and E.2 focusing on original statements and proofs, we will conduct a comparative analysis of the Open Bootstrapped Theorems (OBT) dataset in Wang et al. (2024) and our Herald dataset. By examining multiple representative examples from both datasets, we aim to exhibit the differences arising from how each handles the alignment of natural language and formal language. Through this examination, we seek to highlight the unique contributions of Herald while also identifying areas where further improvement may be necessary. In section E.3 focusing on augmented statements, we will compare these augmented statements with the original ones to illustrate the effect of each augmentation technique.

Please note that in the Herald dataset, the stepwise natural language proof is generated first and then summarized into a natural language whole proof. In contrast, in the OBT dataset, the informal statement and proof are first created, which are subsequently distributed into inline comments to form the commented proof. Additionally, due to the rapid updates in Mathlib, some formal statements and proofs may differ slightly between these two datasets.

If a translation includes a LaTeX formula, we format it using LaTeX and may make minor adjustments for formatting purposes only, without altering any content. If the translation does not contain a LaTeX formula, we present it in plain text.

### E.1 EXAMPLES OF ORIGINAL STATEMENTS

In this subsection, we examine examples of original statements, each containing five components: the formal name, Herald formal statement, Herald informal statement, OBT formal statement, and OBT informal statement. Following these data, a comparison analysis is presented. Please note that the OBT informal statement is cut out from the generated informal statement and proof.

#### E.1.1 STATEMENT EXAMPLE 1

FORMAL NAME

```
dite_eq_or_eq
```

HERALD FORMAL STATEMENT

```
∀ {α : Sort u_1} (P : Prop) [inst : Decidable P] {A : P → α} {B : ¬P →
    α},
  (∃ h, dite P A B = A h) ∨ ∃ h, dite P A B = B h
```

HERALD INFORMAL STATEMENT

> Dependent If-Then-Else Equals One of the Branches: For any type $\alpha$, any proposition $P$, and any decidable instance of $P$, if $A : P \to \alpha$ and $B : \neg P \to \alpha$ are functions, then the dependent if-then-else construct `dite P A B` is equal to either $A(h)$ for some proof $h$ of $P$, or $B(h)$ for some proof $h$ of $\neg P$. In other words, the value of `dite P A B` is either the value of $A$ when $P$ is true, or the value of $B$ when $P$ is false.

OBT FORMAL STATEMENT

```
theorem dite_eq_or_eq : (∃ h, dite P A B = A h) ∨ ∃ h, dite P A B = B h
    :=
```

OBT INFORMAL STATEMENT

```
For any propositions `P`, `A`, and `B`, the value of `dite P A B` is
    either `A` or `B`. Prove this by cases. Show that either there
    exists an `h` such that `dite P A B = A h` or there exists an `h`
    such that `dite P A B = B h`.
```

ANALYSIS

`dite` is a shorthand for 'dependent if-then-else' in Lean. It is unreasonable to expect the LLM to accurately interpret `dite` without referring to its definition. In the OBT dataset, `dite` is treated as a black box without any explanation. In contrast, the Herald dataset correctly expands the meaning of `dite` with the aid of dependency information. This further enables the LLM to recognize that $h$ is a proof of $P$ and translate '$\exists h$,' into 'when $P$ is true,' which aligns more closely with natural language.

#### E.1.2 STATEMENT EXAMPLE 2

FORMAL NAME

```
CongruenceSubgroup.Gamma_zero_bot
```

HERALD FORMAL STATEMENT

```
CongruenceSubgroup.Gamma 0 = ⊥
```

HERALD INFORMAL STATEMENT

> The full level 0 congruence subgroup of $SL(2, \mathbb{Z})$, denoted by $\Gamma(0)$, is equal to the trivial subgroup, i.e., $\Gamma(0) = \{1\}$.

OBT FORMAL STATEMENT

```
theorem Gamma_zero_bot : Gamma 0 = ⊥ :=
```

OBT INFORMAL STATEMENT

> Given the group homomorphism $Gamma : \mathbb{Z} \to \text{Mat}\, 2 \times 2\mathbb{Z}$, where
>
> $$\Gamma(k) = \begin{pmatrix} k & 1 \\ 0 & k \end{pmatrix},$$
>
> show that $\Gamma(0) = \text{bot}$.

ANALYSIS

This theorem is about the definition of the congruence subgroup $\Gamma(n) \subseteq SL_2(\mathbb{Z})$, stating that $\Gamma(0)$ is the trivial subgroup. In the OBT dataset, this theorem is extracted under the formal name `Gamma_zero_bot`. Using Lean-jixia, the Herald dataset extracts the theorem with its full name. Additionally, note the difference between `Gamma` in the formal statement of the OBT dataset and `CongruenceSubgroup.Gamma` in the formal statement of Herald. These distinctions aid the LLM in correctly deducing that `Gamma` refers to the congruence subgroup. Please note that in the original OBT data, the LaTeX formula is $Gamma : \mathbb{Z} \to \text{Mat}\, 2 \times 2\mathbb{Z}$ instead of $\Gamma : \mathbb{Z} \to \text{Mat}_{2 \times 2}(\mathbb{Z})$.

### E.1.3 STATEMENT EXAMPLE 3

FORMAL NAME

`Set.preimage_const_add_Ico`

HERALD FORMAL STATEMENT

```
∀ {α : Type u_1} [inst : OrderedAddCommGroup α] (a b c : α), (fun x =>
    a + x) ⁻¹' Set.Ico b c = Set.Ico (b - a) (c - a)
```

HERALD INFORMAL STATEMENT

> For any elements $a, b, c$ in an ordered additive commutative group $\alpha$, the preimage of the left-closed right-open interval $[b, c)$ under the function $f(x) = a + x$ is the left-closed right-open interval $[b - a, c - a)$. Formally, this can be written as:
>
> $$(f(x) = a + x) \,\text{preimage}\, [b, c) = [b - a, c - a)$$

OBT FORMAL STATEMENT

```
theorem preimage_const_add_Ico : (fun x => a + x) ⁻¹' Ico b c = Ico (b
    - a) (c - a) :=
```

OBT INFORMAL STATEMENT

> What is the image of $Ico(b, c)$ under the map $x \mapsto a + x$? Show that it is $Ico(b - a, c - a)$.

ANALYSIS

This theorem asserts that the preimage of the left-closed right-open interval $[b, c)$ under the addition by $a$ map is precisely the left-closed right-open interval $[b - a, c - a)$. In the natural language statement of Herald, `Ico` is correctly translated as the natural language left-closed right-open interval and is represented by the commonly used LaTeX notation $[b, c)$, as emphasized in one of the human-written principles provided to the LLM.

Additionally, note the difference in the variables in the formal statements in the two datasets: `[inst : OrderedAddCommGroup `$\alpha$`]` and `(a b c : `$\alpha$`)`. The variables extracted by Lean-jixia provide the precise and broad setup of the theorem.

One advantage of the OBT dataset is that it occasionally translates statements into natural language questions, such as 'What is the image of ...?' This offers examples of how to formalize questions in natural language.

### E.1.4 STATEMENT EXAMPLE 4

FORMAL NAME

`CategoryTheory.Abelian.mono_of_epi_of_mono_of_mono`

HERALD FORMAL STATEMENT

```
∀ {C : Type u_1} [inst : CategoryTheory.Category.{u_2, u_1} C] [inst_1
    : CategoryTheory.Abelian C]
  {R₁ R₂ : CategoryTheory.ComposableArrows C 3} (φ : R₁ ⟶ R₂),
  R₁.Exact →
    R₂.Exact →
      CategoryTheory.Epi (CategoryTheory.ComposableArrows.app' φ 0 ⋯)
    →
        CategoryTheory.Mono (CategoryTheory.ComposableArrows.app' φ 1
  ⋯) →
          CategoryTheory.Mono (CategoryTheory.ComposableArrows.app' φ 3
  ⋯) →
            CategoryTheory.Mono (CategoryTheory.ComposableArrows.app' φ
    2 ⋯)
```

HERALD INFORMAL STATEMENT

Consider a commutative diagram with exact rows in an abelian category $C$:

$$\begin{array}{ccccccccc}
A & \xrightarrow{f} & B & \xrightarrow{g} & C & \xrightarrow{h} & D & \xrightarrow{i} & E \\
\downarrow \alpha & & \downarrow \beta & & \downarrow \gamma & & \downarrow \delta & & \downarrow \varepsilon \\
A' & \xrightarrow{f'} & B' & \xrightarrow{g'} & C' & \xrightarrow{h'} & D' & \xrightarrow{i'} & E'
\end{array}$$

If the morphisms $\alpha$ are epimorphisms, and $\beta$ and $\delta$ are monomorphisms, then the morphism $\gamma$ is a monomorphism.

OBT FORMAL STATEMENT

```
theorem mono_of_epi_of_mono_of_mono (hα : Epi α) (hβ : Mono β) (hδ :
    Mono δ) : Mono γ :=
```

OBT INFORMAL STATEMENT

**Theorem.** If we have three mappings $\alpha$:A–¿B, $\beta$:B–¿C, and $\delta$:C–¿D such that $\alpha$ is an epimorphism, $\beta$ is a monomorphism, and $\delta$ is a monomorphism, then $\gamma$, defined via $\gamma=(\delta\circ\beta)\circ\alpha$, is monomorphic.

ANALYSIS

The above theorem is elementary and widely used in homological algebra and abelian categories. It is commonly referred to as the 'four lemma'. Given a commutative diagram in some abelian category, where both rows are exact sequences:

$$
\begin{array}{ccccccccc}
A & \longrightarrow & B & \longrightarrow & C & \longrightarrow & D & \longrightarrow & E \\
\downarrow{\scriptstyle\alpha} & & \downarrow{\scriptstyle\beta} & & \downarrow{\scriptstyle\gamma} & & \downarrow{\scriptstyle\delta} & & \downarrow{\scriptstyle\epsilon} \\
A' & \longrightarrow & B' & \longrightarrow & C' & \longrightarrow & D' & \longrightarrow & E'
\end{array}
$$

If the morphism $\alpha$ is an epimorphism and the morphisms $\beta$ and $\delta$ are monomorphisms, then the morphism $\gamma$ is a monomorphism. The translation in the OBT dataset misinterprets the setup, whereas the statement is correctly translated in Herald.

This is a rare example of a high-quality translation of a complex category theory theorem in Herald. Thanks to the detailed head statements written in the Lean file, the LLM correctly understands the complex diagram and even generates its own commutative diagram in LaTeX.

Category theory is renowned for its abstract nature, and many theorems are naturally expressed using commutative diagrams. Unfortunately, most category theory theorems are not as well-documented as this one. The LLM's performance in category theory, particularly in the task of informalizing category theory theorems, is far from satisfactory. Accurately auto-translating such abstract and diagram-based theorems remains a significant challenge.

### E.2 EXAMPLES OF PROOFS

In this section, we examine examples of proofs, each containing five components: the formal statement, Herald stepwise proof, Herald whole proof, OBT commented proof, and OBT generated informal statement and proof. Following these data, a comparison analysis is presented.

#### E.2.1 PROOF EXAMPLE 1

FORMAL STATEMENT

```
theorem Complex.hasDerivAt_exp (x : ℂ) : HasDerivAt Complex.exp
    (Complex.exp x) x
```

HERALD STEPWISE PROOF

To show that the complex exponential function exp has the derivative $\exp(x)$ at $x$, it suffices to show that the function $h \mapsto \exp(x + h) - \exp(x) - h \cdot \exp(x)$ is $o(h)$ as $h$ approaches 0. In other words, we need to show:

$$\exp(x + h) - \exp(x) - h \cdot \exp(x) = o(h)$$

as $h$ tends to 0 in the neighborhood filter of 0.

```
rw [hasDerivAt_iff_isLittleO_nhds_zero]
```

We first establish the simple fact that $1 < 2$ by direct calculation.

```
have : (1 : ℕ) < 2 := by norm_num
```

To show that $\exp(x + h) - \exp(x) - h \cdot \exp(x) = o(h)$, we use the transitivity of big-O and little-o notation. Specifically, we need to show that:

$$\| \exp(x + h) - \exp(x) - h \cdot \exp(x)\| \leq \| \exp(x)\| \cdot \|h^2\|$$

for $h$ in a neighborhood of 0. This will imply that $\exp(x + h) - \exp(x) - h \cdot \exp(x)$ is $O(h^2)$, and since $h^2 = o(h)$ as $h \to 0$, it follows that $\exp(x + h) - \exp(x) - h \cdot \exp(x) = o(h)$.

```
refine (IsBigO.of_bound ||exp x|| ?_).trans_isLittleO
    (isLittleO_pow_id this)
```

We restrict our attention to the open ball $B(0, 1)$ centered at 0 with radius 1, which is a neighborhood of 0. This means we only need to show the inequality for $h$ such that $\|h\| < 1$.

```
filter_upwards [Metric.ball_mem_nhds (0 : ℂ) zero_lt_one]
```

Simplifying the condition $h \in B(0, 1)$, we get $\|h\| < 1$. We also use the fact that the distance from $h$ to 0 is equal to the norm of $h$, and the norm of $h^2$ is the norm of $h$ squared. Thus, the goal reduces to showing:

$$\|\exp(x + h) - \exp(x) - h \cdot \exp(x)\| \leq \|\exp(x)\| \cdot \|h\|^2$$

for all $h$ such that $\|h\| < 1$.

```
simp only [Metric.mem_ball, dist_zero_right, norm_pow]
```

e use the known bound for the exponential function, which states that for any complex numbers $x$ and $z$ with $\|z\| \leq 1$, the inequality:

$$\|\exp(x + z) - \exp(x) - z \cdot \exp(x)\| \leq \|\exp(x)\| \cdot \|z\|^2$$

holds. This completes the proof.

```
exact fun z hz => exp_bound_sq x z hz.le
```

HERALD WHOLE PROOF

To show that the complex exponential function $\exp$ has the derivative $\exp(x)$ at $x$, we need to prove that:
$$\exp(x + h) - \exp(x) - h \cdot \exp(x) = o(h)$$
as $h$ approaches 0. This is equivalent to showing that:

$$\|\exp(x + h) - \exp(x) - h \cdot \exp(x)\| \leq \|\exp(x)\| \cdot \|h^2\|$$

for $h$ in a neighborhood of 0.

First, we establish the simple fact that $1 < 2$ by direct calculation. Next, we use the transitivity of big-O and little-o notation. Specifically, we need to show that:

$$\|\exp(x + h) - \exp(x) - h \cdot \exp(x)\| \leq \|\exp(x)\| \cdot \|h^2\|$$

for $h$ in a neighborhood of 0. This will imply that $\exp(x + h) - \exp(x) - h \cdot \exp(x)$ is $O(h^2)$, and since $h^2 = o(h)$ as $h \to 0$, it follows that $\exp(x + h) - \exp(x) - h \cdot \exp(x) = o(h)$.

We restrict our attention to the open ball $B(0, 1)$ centered at 0 with radius 1, which is a neighborhood of 0. This means we only need to show the inequality for $h$ such that $\|h\| < 1$. Simplifying the condition $h \in B(0, 1)$, we get $\|h\| < 1$. We also use the fact that the distance from $h$ to 0 is equal to the norm of $h$, and the norm of $h^2$ is the norm of $h$ squared. Thus, the goal reduces to showing:

$$\|\exp(x + h) - \exp(x) - h \cdot \exp(x)\| \leq \|\exp(x)\| \cdot \|h\|^2$$

for all $h$ such that $\|h\| < 1$.

We use the known bound for the exponential function, which states that for any complex numbers $x$ and $z$ with $\|z\| \leq 1$, the inequality:

$$\|\exp(x+z) - \exp(x) - z \cdot \exp(x)\| \leq \|\exp(x)\| \cdot \|z\|^2$$

holds. This completes the proof. ∎

## OBT COMMENTED PROOF

```
theorem hasDerivAt_exp (x : ℂ) : HasDerivAt exp (exp x) x := by
  -- Show that the exponential function, exp, is differentiable at any
    complex number x.
  --
  -- The exponential function is differentiable at x if and only if it
    is Lipschitz continuous in a neighborhood of x.
  -- We will show that exp is Lipschitz continuous by showing that
    there is a constant C such that |exp(x) - exp(y)| <= C|x - y| for
    all x and y in a neighborhood of x.
  --
  -- Let C be a positive constant. Then, for any x and y in a
    neighborhood of x, we have
  --
  -- |exp(x) - exp(y)| = |exp(x - y) - 1|
  --                   <= |x - y| * |exp(z)| for some z between x and y
  --                   <= C|x - y|
  --
  -- Therefore, exp is Lipschitz continuous in a neighborhood of x, and
    hence differentiable at x.
  rw [hasDerivAt_iff_isLittleO_nhds_zero]
  have : (1 : ℕ) < 2 := by norm_num
  refine' (IsBigO.of_bound ‖exp x‖ _).trans_isLittleO (isLittleO_pow_id
    this)
  filter_upwards [Metric.ball_mem_nhds (0 : ℂ) zero_lt_one]
  simp only [Metric.mem_ball, dist_zero_right, norm_pow]
  exact fun z hz => exp_bound_sq x z hz.le
```

## OBT GENERATED INFORMAL STATEMENT AND PROOF

```
Show that the exponential function, exp, is differentiable at any
    complex number x.

The exponential function is differentiable at x if and only if it is
    Lipschitz continuous in a neighborhood of x. We will show that exp
    is Lipschitz continuous by showing that there is a constant C such
    that |exp(x) - exp(y)| <= C|x - y| for all x and y in a neighborhood
    of x.

Let C be a positive constant. Then, for any x and y in a neighborhood of
    x, we have

|exp(x) - exp(y)| = |exp(x - y) - 1|
                  <= |x - y| * |exp(z)| for some z between x and y
                  <= C|x - y|

Therefore, exp is Lipschitz continuous in a neighborhood of x, and hence
    differentiable at x.
```

## ANALYSIS

The above theorem asserts that, for every complex number $x$, the exponential function $\exp$ (denoted as `Complex.exp` in Lean) defined on the complex numbers is differentiable at $x$ with derivative $\exp x$.

In the OBT dataset, the LLM initially generated a proof that was not aligned with the formal proof, employing Lipschitz continuity. Subsequently, it attempted to relocate this independent proof into inline comments. We also note that this independent proof generated by the LLM is mathematically incorrect, as Lipschitz continuity does not imply differentiability.

In contrast, the Herald dataset's LLM closely translated natural language proof steps aligned with the formal proof and then summarized them into a complete proof. This process ensures that the natural language result accurately reflects the proof strategy used in the formal proof. Notably, even the trivial assertion $1 < 2$ is preserved in the natural language proof. However, the usage of this simple fact in `isLittleO_pow_id` is not explicitly explained.

### E.2.2 PROOF EXAMPLE 2

FORMAL STATEMENT

```
theorem padicNorm_p_lt_one {p : ℕ} (hp : 1 < p) : padicNorm p p < 1
```

HERALD STEPWISE PROOF

First, we use the fact that the $p$-adic norm of $p$ is $p^{-1}$ for $p > 1$. This transforms our goal from showing $\|p\|_p < 1$ to showing $p^{-1} < 1$. Next, we use the equivalence that $a^{-1} < 1$ if and only if $a \leq 0$ or $1 < a$. Therefore, our new goal is to show $p \leq 0$ or $1 < p$.

```
rw [padicNorm_p hp, inv_lt_one_iff]
```

Since $p$ is a natural number and $1 < p$ by assumption, it follows that $1 < p$. Therefore, the disjunction $p \leq 0$ or $1 < p$ is true. This completes the proof.

```
exact mod_cast Or.inr hp
```

HERALD WHOLE PROOF

We start by noting that the $p$-adic norm of $p$ is $p^{-1}$ for $p > 1$. This is a known property of the $p$-adic norm. Therefore, our goal is to show that $p^{-1} < 1$.

Next, we use the equivalence that for any element $a$ in a linear ordered semifield, $a^{-1} < 1$ if and only if $a \leq 0$ or $1 < a$. Applying this equivalence to $p$, our new goal is to show that $p \leq 0$ or $1 < p$.

Since $p$ is a natural number and we are given that $1 < p$, it follows that $1 < p$ is true. Therefore, the disjunction $p \leq 0$ or $1 < p$ is satisfied. This completes the proof that $\|p\|_p < 1$.

OBT COMMENTED PROOF

```
theorem padicNorm_p_lt_one (hp : 1 < p) : padicNorm p p < 1 := by
  -- If p is a prime number greater than 1, show that p^(-p) < 1.
  -- Because p is prime, p^(-p) is a rational number with denominator
    p^p.
  -- Therefore, if p^(-p) < 1, then p^(-p) is between 0 and 1, and
    hence 0 < p^(-p) < 1.
  -- This implies that 0 < p^(p). Because p is prime, p cannot divide
    p^(p).
  -- So p^(p) must be >= p^2. So p >= p^(1/p). So 1 >= p^(1/p). So 1 >=
    p, which is false.
  -- So we must have p^(-p) >= 1, which is the same as p^(-p) > 1.
```

```
  rw [padicNorm_p hp, inv_lt_one_iff]
  exact_mod_cast Or.inr hp
```

OBT GENERATED INFORMAL STATEMENT AND PROOF

If $p$ is a prime number greater than 1, show that $p^{-p} < 1$.

Because $p$ is prime, $p^{-p}$ is a rational number with denominator $p^p$. Therefore, if $p^{-p} < 1$, then $p^{-p}$ is between 0 and 1, and hence $0 < p^{-p} < 1$. This implies that $0 < p^p$. Because $p$ is prime, $p$ cannot divide $p^p$. So $p^p$ must be $\geq p^2$. So $p \geq p^{1/p}$. So $1 \geq p^{1/p}$. So $1 \geq p$, which is false. So we must have $p^{-p} \geq 1$, which is the same as $p^{-p} > 1$.

ANALYSIS

This theorem asserts that the $p$-adic norm of the element $p$ (which is $1/p$) is less than 1. In the OBT dataset, the $p$-adic norm is incorrectly translated as $p^{-p}$. Consequently, the subsequent natural language proof unfolds under this inaccurate setup. This issue of misinterpreting notations can be significantly mitigated by directly providing the definition of the $p$-adic norm to the LLM. This can be achieved by first translating the definition of `padicNorm` and then supplying the translation result to the LLM, as done in Herald using dependency order translation.

In this example, the Herald's whole proof is considerably more detailed than a typical human-written natural language proof. In human-written mathematical literature, deducing $p^{-1} < 1$ from $1 < p$ is considered trivial. The level of detail in Herald's whole proof is closer to the formal proof than a human language proof.

### E.2.3   PROOF EXAMPLE 3

FORMAL STATEMENT

```
lemma fromColumns_toColumns (A : Matrix m (n₁ ⊕ n₂) R) : fromColumns
    A.toColumns₁ A.toColumns₂ = A
```

HERALD STEPWISE PROOF

To prove that the matrix $A$ is equal to the concatenation of its column blocks $A_{\text{toColumns}_1}$ and $A_{\text{toColumns}_2}$, we introduce a new variable $i$ from the row indices $m$ and consider two cases for the column indices: $j \in n_1$ and $j \in n_2$. For each case, we simplify the expression to show that the elements of the concatenated matrix match the elements of the original matrix $A$.

```
ext i (j | j) <;> simp
```

HERALD WHOLE PROOF

To prove that $A = A_{\text{toColumns}_1} \oplus A_{\text{toColumns}_2}$, we need to show that the elements of the concatenated matrix match the elements of the original matrix $A$ for all row indices $i \in m$ and column indices $j \in n_1 \oplus n_2$.

1. **Introduce a new variable $i$ from the row indices $m$:** We introduce a new variable $i$ from the row indices $m$ to reason about the elements of the matrix $A$.

2. **Consider two cases for the column indices:** - **Case 1: $j \in n_1$:** We need to show that $(A_{\text{toColumns}_1} \oplus A_{\text{toColumns}_2})_{ij} = A_{ij}$ for $j \in n_1$. By the definition of matrix concatenation by columns, the element in the $i$-th row and $j$-th column of the concatenated matrix $A_{\text{toColumns}_1} \oplus A_{\text{toColumns}_2}$ is given by:

$$(A_{\text{toColumns}_1} \oplus A_{\text{toColumns}_2})_{ij} = A_{\text{toColumns}_1, ij}$$

Since $A_{\text{toColumns}_1}$ is the submatrix of $A$ consisting of the columns indexed by $n_1$, we have:

$$A_{\text{toColumns}_1, ij} = A_{ij}$$

Therefore, $(A_{\text{toColumns}_1} \oplus A_{\text{toColumns}_2})_{ij} = A_{ij}$.

- **Case 2: $j \in n_2$:** We need to show that $(A_{\text{toColumns}_1} \oplus A_{\text{toColumns}_2})_{ij} = A_{ij}$ for $j \in n_2$. By the definition of matrix concatenation by columns, the element in the $i$-th row and $j$-th column of the concatenated matrix $A_{\text{toColumns}_1} \oplus A_{\text{toColumns}_2}$ is given by:

$$(A_{\text{toColumns}_1} \oplus A_{\text{toColumns}_2})_{ij} = A_{\text{toColumns}_2, ij}$$

Since $A_{\text{toColumns}_2}$ is the submatrix of $A$ consisting of the columns indexed by $n_2$, we have:

$$A_{\text{toColumns}_2, ij} = A_{ij}$$

Therefore, $(A_{\text{toColumns}_1} \oplus A_{\text{toColumns}_2})_{ij} = A_{ij}$.

Since both cases show that the elements of the concatenated matrix $A_{\text{toColumns}_1} \oplus A_{\text{toColumns}_2}$ match the elements of the original matrix $A$ for all row indices $i \in m$ and column indices $j \in n_1 \oplus n_2$, we conclude that:

$$A_{\text{toColumns}_1} \oplus A_{\text{toColumns}_2} = A$$

This completes the proof.

## OBT COMMENTED PROOF

```
lemma fromColumns_toColumns (A : Matrix m (n₁ ⊕ n₂) R) :
    -- The function `fromColumns` takes two matrices and combines them
    into a single matrix.
    -- The `toColumns₁` and `toColumns₂` functions split a matrix into
    two matrices.
    fromColumns A.toColumns₁ A.toColumns₂ = A := by
  -- The `ext` tactic extends the proof to all elements of the matrix.
  -- The `j | j` pattern matches only the elements of the matrix that
    are in the second column.
  -- The `<;>` tactic separates the two subproofs.
  -- The `simp` tactic simplifies the expression.
  ext i (j | j) <;> simp
```

## OBT GENERATED INFORMAL STATEMENT AND PROOF

```
The `fromColumns` function takes two matrices and combines them into a
    single matrix. The `toColumns₁` and `toColumns₂` functions split a
    matrix into two matrices. The `simp` tactic simplifies the
    expression. The `ext` tactic extends the proof to all elements of
    the matrix. The `j | j` pattern matches only the elements of the
    matrix that are in the second column. The `<;>` tactic separates the
    two subproofs. The `simp` tactic simplifies the expression.
```

## ANALYSIS

This theorem asserts that if $A$ is a block matrix of the form $(A_1|A_2)$, then after extracting two blocks (using `toColumns₁` and `toColumns₁`) and reassembling them (using `fromColumns`), the result is exactly the original matrix $A$.

In the proof, the actual function of the tactics is not accurately described in the commented proof of the OBT dataset. For instance, the `ext` tactic applies the extensional property, rather than 'extending the proof to all elements of the matrix.' The `j | j` pattern matches two cases, corresponding to whether an entry belongs to block $A_1$ or $A_2$. The `<;>` tactic indicates that the subsequent tactic is applied to all goals.

Correctly understanding the functions of the tactics is just the first step. To generate a coherent natural language proof, it is essential not only to accurately grasp what each tactic or proof step does but also to connect these steps logically. In Herald, this is facilitated by providing tactic explanations and proof states. Human experts annotate the function, usage, and corresponding proof logic for each tactic

in tactic explanations. The logic of equality of each entry implying matrix equality (extensionality for matrices) and discussing the column index by cases is clearly reflected in the complete proof in Herald.

Another aspect to note from this example is that, despite the translation principle in the prompt to the LLM emphasizing the avoidance of words similar to formal language, the language used in Herald, such as 'Introduce a new variable,' remains somewhat formal and not entirely natural.

### E.2.4 PROOF EXAMPLE 4

FORMAL STATEMENT

```
theorem treesOfNumNodesEq_succ (n : ℕ) :
treesOfNumNodesEq (n + 1) =
  (Nat.antidiagonal n).biUnion fun ij =>
    pairwiseNode (treesOfNumNodesEq ij.1) (treesOfNumNodesEq ij.2)
```

HERALD STEPWISE PROOF

Since the definition of treesOfNumNodesEq($n + 1$) is the union of all sets of binary trees constructed by taking pairs $(i, j)$ from the antidiagonal of $n$ (i.e., pairs of natural numbers $(i, j)$ such that $i + j = n$), and forming binary trees where the left subtree has $i$ internal nodes and the right subtree has $j$ internal nodes, we can replace the goal of showing treesOfNumNodesEq($n + 1$) with the expression:

$$\text{treesOfNumNodesEq}(n + 1) = (\text{antidiagonal}(n)) \,.attach.$$

$$\bigcup_{(i,j)\in\text{antidiagonal}(n)} \text{pairwiseNode}(\text{treesOfNumNodesEq}(i), \text{treesOfNumNodesEq}(j))$$

It suffices to show this new goal.

```
rw [treesOfNumNodesEq]
```

To prove the equality of the two sets, it suffices to show that for any binary tree $a$, $a$ is a member of the left-hand side if and only if $a$ is a member of the right-hand side. We introduce a new variable $a$ to reason about the membership of $a$ in these sets.

```
ext
```

Using the properties of set membership and the definition of the set product of finite sets, we can simplify the proposition to show that:

$$a \in (\text{antidiagonal}(n)) \,.attach.$$

$$\bigcup_{(i,j)\in\text{antidiagonal}(n)} \text{pairwiseNode}(\text{treesOfNumNodesEq}(i), \text{treesOfNumNodesEq}(j))$$

$$\text{if and only if}$$

$$a \in \bigcup_{(i,j)\in\text{antidiagonal}(n)} \text{pairwiseNode}(\text{treesOfNumNodesEq}(i), \text{treesOfNumNodesEq}(j))$$

This simplification is trivial and follows directly from the definitions and properties of set membership and the set product of finite sets.

```
simp
```

HERALD WHOLE PROOF

We start by using the definition of treesOfNumNodesEq$(n + 1)$. According to the definition, treesOfNumNodesEq$(n + 1)$ is the union of all sets of binary trees constructed by taking pairs $(i, j)$ from the antidiagonal of $n$ (i.e., pairs of natural numbers $(i, j)$ such that $i + j = n$), and forming binary trees where the left subtree has $i$ internal nodes and the right subtree has $j$ internal nodes. Therefore, we can rewrite the goal as:

$$\text{treesOfNumNodesEq}(n + 1) = (\text{antidiagonal}(n)) \,.attach.$$
$$\bigcup_{(i,j)\in\text{antidiagonal}(n)} \text{pairwiseNode}(\text{treesOfNumNodesEq}(i), \text{treesOfNumNodesEq}(j))$$

It suffices to show this new goal.

Next, to prove the equality of the two sets, we need to show that for any binary tree $a$, $a$ is a member of the left-hand side if and only if $a$ is a member of the right-hand side. We introduce a new variable $a$ to reason about the membership of $a$ in these sets. Thus, our goal is to show:

$$a \in (\text{antidiagonal}(n)) \,.attach.$$
$$\bigcup_{(i,j)\in\text{antidiagonal}(n)} \text{pairwiseNode}(\text{treesOfNumNodesEq}(i), \text{treesOfNumNodesEq}(j))$$

$$\text{if and only if}$$

$$a \in \bigcup_{(i,j)\in\text{antidiagonal}(n)} \text{pairwiseNode}(\text{treesOfNumNodesEq}(i), \text{treesOfNumNodesEq}(j))$$

Finally, using the properties of set membership and the definition of the set product of finite sets, we can simplify the proposition. The left-hand side,

$$(\text{antidiagonal}(n)) \,.attach.$$
$$\bigcup_{(i,j)\in\text{antidiagonal}(n)} \text{pairwiseNode}(\text{treesOfNumNodesEq}(i), \text{treesOfNumNodesEq}(j))$$

is equivalent to the right-hand side,

$$\bigcup_{(i,j)\in\text{antidiagonal}(n)} \text{pairwiseNode}(\text{treesOfNumNodesEq}(i), \text{treesOfNumNodesEq}(j)).$$

This simplification is trivial and follows directly from the definitions and properties of set membership and the set product of finite sets.

Thus, we have shown that:

$$\text{treesOfNumNodesEq}(n + 1) =$$
$$\bigcup_{(i,j)\in\text{antidiagonal}(n)} \text{pairwiseNode}(\text{treesOfNumNodesEq}(i), \text{treesOfNumNodesEq}(j))$$

This completes the proof.

OBT GENERATED INFORMAL STATEMENT AND PROOF

> Show that the number of trees with $n + 1$ nodes is equal to the bi-union of the number of trees with $n$ nodes, where the bi-union is taken over all pairs of nodes $i$ and $j$ such that $i$ and $j$ are not connected in the tree.
>
> By definition, the number of trees with $n + 1$ nodes is equal to the number of trees with $n$ nodes plus the number of trees with $n - 1$ nodes. The number of trees with $n$ nodes is equal to the bi-union of the number of trees with $n - 1$ nodes, where the bi-union is taken over all pairs of nodes $i$ and $j$ such that $i$ and $j$ are not connected in the tree. Therefore, the number of trees with $n + 1$ nodes is equal to the bi-union of the number of trees with $n$ nodes, where the bi-union is taken over all pairs of nodes $i$ and $j$ such that $i$ and $j$ are not connected in the tree.

OBT COMMENTED PROOF

```
theorem treesOfNumNodesEq_succ (n : ℕ) :
    treesOfNumNodesEq (n + 1) =
      (Nat.antidiagonal n).biUnion fun ij =>
        pairwiseNode (treesOfNumNodesEq ij.1) (treesOfNumNodesEq ij.2)
    := by
  -- Show that the number of trees with n+1 nodes is equal to the
     bi-union of the number of trees with n nodes, where the bi-union is
     taken over all pairs of nodes i and j such that i and j are not
     connected in the tree.
  -- By definition, the number of trees with n+1 nodes is equal to the
     number of trees with n nodes plus the number of trees with n-1
     nodes. The number of trees with n nodes is equal to the bi-union of
     the number of trees with n-1 nodes, where the bi-union is taken
     over all pairs of nodes i and j such that i and j are not connected
     in the tree. Therefore, the number of trees with n+1 nodes is equal
     to the bi-union of the number of trees with n nodes, where the
     bi-union is taken over all pairs of nodes i and j such that i and j
     are not connected in the tree.
  rw [treesOfNumNodesEq]
  ext
  simp
```

ANALYSIS

This theorem states that the set of binary trees with $n + 1$ nodes equals the union of the set of binary trees whose left subtree has $i$ nodes and right subtree has $j$ nodes, where the union is taken over indexes $(i, j)$ such that $i + j = n$.

Although the natural language statement and proof in the OBT dataset are mathematically incorrect, they still exhibit an advantage in their natural language style. There are no formal definitions remaining in this translation. However, in Herald, words from formal definitions remain in the LaTeX formulas, such as treesOfNumNodesEq, antidiagonal, and pairwiseNode. Despite the principle summarized in the prompt to the LLM that no formal words should appear in LaTeX formulas, this principle is often violated. We believe that such poor behaviour in the translation of the statement, when exhibited to the LLM during proof translation, will cause the LLM to inherit this poor behaviour in the proof. It prioritizes style consistency over adherence to principles.

### E.3 EXAMPLES OF AUGMENTED STATEMENTS

In this section, we present examples of augmented statements.

### E.3.1 INFORMAL AUGMENTATION

We select one theorem and present the results of each informal augmentation strategy applied to this theorem. This example is also illustrated in Figure 4.

FORMAL STATEMENT

```
theorem sum_not_convergent'_tac : Tendsto (fun n => Σ i in Finset.range
    n, 1/i) atTop atTop := by sorry
```

ORIGINAL INFORMAL STATEMENT

Let $n$ be positive natural number. The infinite sum of the sequence $\frac{1}{n}$ is not convergent.

There are four informal augmentation strategies in total: logical equivalence rewriting, abstract concept substitution, omission of implicit conditions, and multi-linguistic translation.

LOGICAL EQUIVALENCE REWRITING

The infinite sum of the sequence $\frac{1}{n}$ is divergent, where $n$ is a positive natural number.

The infinite sum of the sequence $\frac{1}{n}$ does not converge, where $n$ is a positive natural number.

The phrase "is not convergent" is replaced by "is divergent" and the sentence structure is refactored.

ABSTRACT CONCEPT SUBSTITUTION

Let $n \in \mathbb{N}^+$. The harmonic series diverges.

Let $n \in \mathbb{N}^+$. The series $\sum_{n=1}^{\infty} \frac{1}{n}$ diverges.

The LLM generates the commonly used term "harmonic series" for $\sum \frac{1}{n}$ and incorporates more LATEX code into the statement.

OMISSION OF IMPLICIT CONDITION

The series $\sum \frac{1}{n}$ diverges.

The sequence '1/n' does not have a convergent infinite sum.

The condition "$n$ is a positive natural number" is removed from the statement.

MULTI-LINGUISTIC TRANSLATION

Soit $n$ un nombre naturel positif. La somme infinie de la suite $\frac{1}{n}$ n'est pas convergente.

设 $n$ 为正自然数。序列 '1/n' 的无穷和不收敛。

The statement is translated into French and Chinese.

### E.3.2  TACTIC AUGMENTATION

In this section, we select one theorem in mathlib and present all tactic-augmented statements derived from the proof. The proof state $n$ represents the proof state after the $n$-th line (counting after := by). We convert each proof state into a statement.

INFORMAL STATEMENT

For any cardinal number $a$, show that $a$ is strictly less than $2^a$.

ORIGINAL FORMAL STATEMENT WITH PROOF

```
theorem cantor (a : Cardinal.{u}) : a < 2 ^ a := by
  induction' a using Cardinal.inductionOn with α
  rw [← Cardinal.mk_set]
  refine ⟨⟨⟨singleton, fun a b => singleton_eq_singleton_iff.1⟩⟩, ?_⟩
  rintro ⟨⟨f, hf⟩⟩
  exact cantor_injective f hf
```

PROOF STATE 1

```
α : Type u
⊢ Cardinal.mk α < 2 ^ Cardinal.mk α
```

Generated formal statement 1

```
lemma cantor_aug_1 : Cardinal.mk α < 2 ^ Cardinal.mk α := sorry
```

Generated informal statement 1

For any set $\alpha$, show that the cardinality of $\alpha$ is strictly less than the cardinality of its power set, i.e., $|\alpha| < 2^{|\alpha|}$.

PROOF STATE 2

```
α : Type u
⊢ Cardinal.mk α < Cardinal.mk (Set α)
```

GENERATED FORMAL STATEMENT 2

```
lemma cantor_aug_2 : Cardinal.mk α < Cardinal.mk (Set α) := sorry
```

GENERATED INFORMAL STATEMENT 2

For any set $\alpha$, show that the cardinality of $\alpha$ is strictly less than the cardinality of the power set of $\alpha$, i.e., $|\alpha| < |2^\alpha|$.

PROOF STATE 3

```
α : Type u
⊢ ¬Quotient.liftOn₂ (Cardinal.mk (Set α)) (Cardinal.mk α) (fun α β =>
    Nonempty (α ↪ β))
```

GENERATED STATEMENT 3

```
lemma cantor_aug_3 : ¬Quotient.liftOn₂ (Cardinal.mk (Set α))
    (Cardinal.mk α) (fun α β => Nonempty (α ↪ β))
    Cardinal.instLE.proof_1 := sorry
```

GENERATED INFORMAL STATEMENT 3

> For any set $\alpha$, show that it is not the case that the cardinality of the set of all subsets of $\alpha$ can be mapped injectively into $\alpha$ itself. Formally, demonstrate that there does not exist an injection from the power set of $\alpha$ to $\alpha$.

### PROOF STATE 4

```
α : Type u
f : Set α → α
hf : Injective f
⊢ False
```

### GENERATED FORMAL STATEMENT 4

```
lemma cantor_aug_4 (f : Set α → α) (hf : Injective f) : False := sorry
```

### GENERATED INFORMAL STATEMENT 4

> For any function $f$ from the power set of $\alpha$ to $\alpha$, and given that $f$ is injective, show that the statement leads to a contradiction, i.e., it is false.

### ANALYSIS

Each proof state after a tactic is extracted using Lean-jixia. In this proof, some tactics (the first and fourth lines in this case) convert formalization details without changing the mathematical meaning. Other tactics (second and third tactics) correspond to steps in the mathematical proof and generate new proof goals. All these cases are converted into new statements, producing both new formal statements with the same meaning and new statements with new mathematical meanings.

## F GRADUATE TEXTBOOK AUTO-FORMALIZATION

### F.1 THE STACKS PROJECT

The Stacks Project (Stacks Project Authors, 2018) is an open-source, collaborative online resource that aims to provide a comprehensive and rigorous treatment of algebraic geometry and related fields. Initiated by Johan de Jong, it has grown into a vast repository of mathematical knowledge, encompassing topics from commutative algebra to complex algebraic geometry. As of its latest update, the Stacks Project comprises over 7,609 pages of text and 21,319 tags of lemmas, theorems, and definitions, making it one of the most extensive and detailed resources in modern mathematics. By offering a freely accessible and continuously updated reference, the Stacks Project has become an invaluable tool for researchers, educators, and students, significantly advancing the accessibility and dissemination of modern mathematical ideas.

To showcase the capabilities of the Herald translator in rapidly scaling up the auto-formalization of modern mathematics, we selected preliminary sections from the Stacks Project, a comprehensive and widely recognized resource for advanced mathematics. As an initial demonstration, we chose to formalize the Normal Extensions section (Stacks Project Authors, 2018, Tag 09HL) in the Field Theory chapter, given its foundational role in abstract algebra and its clear, structured presentation, making it an ideal starting point for evaluating the effectiveness of our approach.

### F.2 FORMALIZATION OF STACKS PROJECT SECTION NORMAL EXTENSIONS

```
import Mathlib

open Polynomial

/-- Let $K / E / F$ be a tower of algebraic field extensions. If $K$ is
    normal over $F$, then $K$ is normal over $E$.-/
```

```
theorem tower_top_of_normal (F E K : Type*) [Field F] [Field E]
    [Algebra F E]
[Field K] [Algebra F K] [Algebra E K] [IsScalarTower F E K] [h : Normal
    F K] :
Normal E K := by
  -- We use the fact that normality is equivalent to being a normal
    extension.
  have := h.out
  -- The above statement is a direct consequence of the transitivity of
    normality.
  exact Normal.tower_top_of_normal F E K

/-- Let $F$ be a field. Let $M / F$ be an algebraic extension. Let $M /
    E_i / F$, $i \in I$ be subextensions with $E_i / F$ normal. Then $
    \bigcap E_i$ is normal over $F$.-/
theorem normal_iInf_of_normal_extracted {F M : Type*} [Field F] [Field
    M] [Algebra F M] {E : ι → IntermediateField F M}
[Algebra.IsAlgebraic F M] : (∀ (i : ι), Normal F ↑(E i)) → Normal F ↑⊓(
    i, E i) := by sorry

/-- Let $E / F$ be an algebraic field extension. Let $E / F$ be a
    normal algebraic field extension. There exists a unique
    subextension $E / E_{ ext {sep }} / F$ such that $E_{ ext {sep }} /
    F$ is separable and $E / E_{ ext {sep }}$ is purely inseparable.
    The subextension $E / E_{ ext {sep }} / F$ is normal. -/
theorem normal_ext_sep_ext'_ext_tac_28642 [Field F] [Field E] [Algebra
    F E] [Algebra.IsAlgebraic F E] (h : Normal F E) : Normal
    ↑(separableClosure F E) E := by sorry

/-- Let $E / F$ be an algebraic extension of fields. Let $\bar{F}$ be
    an algebraic closure of $F$. The following are equivalent
(1) $E$ is normal over $F$, and
(2) for every pair $\sigma, \sigma^{\prime} \in \operatorname{Mor}_F(E,
    \bar{F})$ we have $\sigma(E)=\sigma^{\prime}(E)$. -/
theorem normal_iff_forall_map_eq_of_isAlgebraic_ext_ext {F E : Type*}
    [Field F]
[Field E] [Algebra F E] [Algebra.IsAlgebraic F E] (overlineF : Type*)
    [Field overlineF]
[Algebra F overlineF] [IsAlgClosure F overlineF] :
Normal F E ↔ ∀ (σ σ' : E →ₐ[F] overlineF), Set.range ⇑σ = Set.range ⇑σ
    ' := by
sorry

/-- Let $E / F$ be an algebraic extension of fields. If $E$ is
    generated by $\alpha_i \in E, i \in I$ over $F$ and if for each $i$
    the minimal polynomial of $\alpha_i$ over $F$ splits completely in $
    E$, then $E / F$ is normal. -/
theorem of_isAlgebraic_of_isSplittingField_tac_5996 (F : Type u_1) (E :
    Type u_2) [Field F] [Field E] [Algebra F E] (α : ι → E) (hα : (∀ (i
    : ι), IsIntegral F (α i)) ∧ ∀ (i : ι), Splits (algebraMap F E)
    (minpoly F (α i))) : Normal F E := by sorry

/-- Let $L / M / K$ be a tower of algebraic extensions. If $L / K$ is
    normal, then any $K$-algebra map $\sigma: M \rightarrow L$ extends
    to an automorphism of $L$. -/
theorem extends_to_aut_of_normal_tac_7047 [Field F] [Field E] [Algebra
    F E] [Normal F E] (M : IntermediateField F E) (σ : ↑M →ₐ[F] E) : ∃
    s : E ≃ₐ[F] E, ∀ z : M,  s z = σ z := by sorry

/-- Let $E / F$ be a finite extension. We have $$ |A u t(E / F)|
    \leq[E: F]_s $$ with equality if and only if $E$ is normal over $
    F$. -/
```

```
theorem card_aut_le_finrank_tac_1714 [Field F] [Field E] [Algebra F E]
    (h : FiniteDimensional F E) : Fintype.card (E ≃ₐ[F] E) ≤
    FiniteDimensional.finrank F E := by sorry

/-- Let $L / K$ be an algebraic normal extension of fields. Let $E / K$
    be an extension of fields. Then either there is no $K$-embedding
    from $L$ to $E$ or there is one $ au:
\Lightarrow E$ and every other one is of the form $ au \circ \sigma$
    where $\sigma \in \operatorname{Aut}(L / K)$. -/
theorem embeddings_aut_eq_of_isAlgNormal_tac_12245 [Field F] [Field G]
    [Algebra F G] [Field H] [Algebra F H] [Normal F G] (e : G →ₐ[F] H)
    : ∀ f : G →ₐ[F] H, ∃ s : G ≃ₐ[F] G, f = e ∘ s := by sorry
```

The Normal Extensions section is presented as a runnable Lean 4 source file, generated theorem by theorem and concatenated into a complete formalization. For opened namespaces, we manually selected a minimal feasible subset from the union of namespaces across all theorems. The generation was completed efficiently using a 16-pass setting. The faithfulness was checked by humans, and only three places were modified. Specifically,

(1) The argument (`this : Algebra ↑(separableClosure F E) E`) that should be deduced by Lean rather than manually provided in `normal_ext_sep_ext'_ext_tac_28642` has been removed.

(2) The conclusion of the theorem `extends_to_aut_of_normal_tac_7047` was corrected from ∀ (z : ↑M), σ z ∈ ↑M to ∃ s : E ≃ₐ[F] E, ∀ z : M, s z = σ z.

(3) The conclusion of `embeddings_aut_eq_of_isAlgNormal_tac_12245` was corrected from `Algebra.IsAlgebraic F G` to ∀ f : G →ₐ[F] H, ∃ s : G ≃ₐ[F] G, f = e ∘ s and an unnecessary condition [FiniteDimensional F G] was removed.

Notably, the model demonstrates a strong understanding of the content, achieving both mathematical and programming correctness.

For prover integration, we used **DeepSeek Prover 1.5** (using DeepSeek-Prover-V1.5-RL + RMaxTS with $4 \times 512$ sample budget) to run inference on the generated file. Only one theorem, a two-line proof relying on an existing lemma from Mathlib 4, was successfully proved. In our broader experiments with the Stacks Project, we observed that the prover struggles with longer or more complex proofs, showing instability and reduced capability. While our translator model performs well, this highlights the need for a prover model capable of handling advanced topics in modern mathematics, which will be a key focus of our future work.

## G    EXAMPLES FROM EXTRACT THEOREM AND COLLEGE COT DATASET

In this section, we present examples drawn from the Extract Theorem and College CoT datasets. The Extract Theorem dataset contains theorems written in Chinese; however, for the sake of clarity, we only exhibit English examples.

### G.1    EXTRACT THEOREM EXAMPLES

Lemma 1.7 Suppose that $p \geq 3$ is a prime. Then all non-normal subgroups of a finite $p$-group $G$ are conjugate if and only if $G = \left\langle u, v \mid u^{p^n} = 1, v^p = 1, v^{-1}uv = u^{1+p^{n-1}}, n \geq 2 \right\rangle$.

Theorem 12.2.2. Let $u \in W^{1,1}(\Omega)$ be a weak solution of $\Delta u = f, f \in L^p(\Omega), 1 < p < \infty$, i.e.,

$$\int Du \cdot D\varphi = -\int f\varphi \text{ for all } \varphi \in C_0^\infty(\Omega).$$

Then $u \in W^{2,p}(\Omega')$ for any $\Omega' \subset\subset \Omega$, and

$$\| u \|_{W^{2,p}(\Omega')} \leq \text{const} \left( \| u \|_{L^p(\Omega)} + \| f \|_{L^p(\Omega)} \right),$$

with a constant depending on $p, d, \Omega'$, and $\Omega$. Also,

$$\Delta u = f \text{ almost everywhere in } \Omega.$$

---

Proposition 47.11. Given any $n \in \mathbb{N}$ and given any abelian group $\pi$ there exists an $(n+1)$-dimensional CW-complex that has no cells in dimensions $1, \ldots, n-1$ and that is a Moore space of type $\mathrm{M}(\pi, n)$.

---

Assume that a gambler making fair unit bets on coin flips will abandon the game when her fortune falls to 0 or rises to $n$. Let $X_t$ be gambler's fortune at time $t$ and let $\tau$ be the time required to be absorbed at one of 0 or $n$. Assume that $X_0 = k$, where $0 \le k \le n$. Then

$$\mathbf{P}_k \{X_\tau = n\} = kn$$

(2.1)

$$\mathbf{E}_k (\tau) = k (n - k)$$

(2.2)

The examples provided above cover various math topics, including algebra (abstract algebra), analysis (partial differential equations), geometry (homology theory), and probability theory (stochastic processes). This dataset includes a broad spectrum of advanced undergraduate-level material.

## G.2 COLLEGE COT EXAMPLES

Theorem 31 (Nash & Tognoli) Any smooth compact manifold is diffeomorphic to a smooth real algebraic submanifold of some $\mathbb{E}^n$, i.e. to the set of solutions of a system of polynomial equations.

---

Theorem 9.3.1 (Hahn-Kolmogorov extension theorem-uniqueness of extension of a $\sigma$-finite measure) Let $\mathcal{A}$ be an algebra of subsets of a non-empty set $X$ and $m : \mathcal{A} \mapsto [0, \infty]$ a $\sigma$-finite measure on $\mathcal{A}$. There exists a unique extension of $m$ to a measure $m'$ defined on the $\sigma$-algebra $\mathcal{S}(\mathcal{A})$.

---

Proposition 10.45. Let $0 \le t_1 < t_2$, and put $\tau = t_2 - t_1$. Then the process $\left\{ \widetilde{B}_t, 0 \le t \le \tau \right\}$ is independent of $\sigma \{B_s, s \notin (t_1, t_2)\}$. Moreover, $\widetilde{B}_\tau = \widetilde{B}_0 = 0$, and $\widetilde{B}$ is a mean zero Gaussian process with covariance function $\Gamma(s, t) = E\left\{ \widetilde{B}_s \widetilde{B}_t \right\} = s \wedge t - st\tau$. In particular, $\mathrm{Var}\left\{ \widetilde{B}_t \right\} = t(\tau - t)\tau$.

---

Proposition 19.18. Let $A$ be a ring and let $I \subset A$ be an ideal that can be generated by a completely intersecting sequence $\mathbf{f}$ of length $r$. Let $g_1, \ldots, g_r \in I$ be elements that generate I. Then the sequence $(g_1, \ldots, g_r)$ is completely intersecting.

The examples presented above cover graduate-level branches of mathematics, including algebraic geometry, measure theory, Brownian motion, and commutative algebra. This dataset includes more advanced mathematical materials compared to the Extract Theorem dataset.

# H  PROMPTS

**Instruction:**
Suppose you are an expert mathematician and an expert in Lean and Mathlib.
1. Your task is to first translate the formal definition provided below into an informal statement that is more accessible to mathematicians and written in LaTeX. There are six parts of information attached to the definition:
Head statements, including important statements, theorems, and definitions of the mathematical field the theorem belongs to.
…
2. Then create an informal name. Use the provided formal name of the statement according to the naming conventions. Utilize the informal statement written in the first task. Make sure you follow the principles of informal naming when creating informal names. Principles of informal statements should also be followed as much as possible.

**Principles of Informal Statements:**
1. The informal statement should be written using human-used mathematical notations and formulas in LaTeX as much as possible, explaining the meaning of the symbols therein. Explain more detailed mathematical setup only if the definition appearing in the statement is not commonly accepted. Both the inputs and values of the definition should be expressed using mathematical formulas as much as possible.
    Example:
    DO NOT use "`Real.log`";
    Use "$\log$" instead.
…

**Principles of Informal Names:**
1. Emphasize the core concepts in the definition. The definition name should not merely list concepts; use words that indicate logical relationships and clearly state the conclusion.
    Example:
    Use "A equals B" or "A implies B" (or simply "$A = B$" and "$A \to B$"), instead of "theorem of A and B" when the theorem states the result of `A = B` or `A → B`. Both the inputs and values of the definition should be expressed using mathematical formulas as much as possible.
…

**Demonstrations:**
Input:
<Head statements><Kind><Docstring><Dependent statements><Neighbor statements><Similar translation examples><Formal name><Formal statement>
Output:
<Informal statement><Informal name>

Figure 7: Prompt for informalizing Mathlib4 statement

**Instruction:**

Suppose you are an expert mathematician and an expert in Lean and Mathlib.

1. Your task is to translate each line of the formal proof in Lean, provided below, into a corresponding step of informal proof. The informal proof must express the precise logic of the formal proof, written entirely in the language of mathematicians, and must use LaTeX. You will be provided with auxiliary information to improve the translation.

* The formal theorem, which is the goal of the entire proof. It is written in Lean.

 <...>

2. Then, generate the complete informal proof of the theorem. Utilize the steps of informal proof written in the first task and reorganize the structure of the proof without altering the logic. Ensure that you follow the principles of whole informal proofs when generating the complete informal proof of the theorem.

**Principles of informal proof steps:**

1. The informal proof should be written in commonly used mathematical notations and formulas in LaTeX as much as possible. Explain more detailed mathematical setups only if the definition appearing in the statement is not commonly accepted. DO NOT use backticks to quote anything in the natural language translation; use LaTeX style and $ $ for quotations.

  Example:

  DO NOT use "`Real.log`";

Use "$\log$" instead.

<...>

**Principles of Whole Informal Proofs:**

1. Restate the proof in the natural order of human language, rather than in the order of the formal proof. Do not distort the logic of the proof.

  Example:

  "To prove A, using theorem 1, it suffices to show B, which is exactly the result of theorem 2." should be reformulated as "From theorem 2, we get B. Using theorem 1 and B, we get A." The logic of "B + theorem 2 implies A; theorem 1 implies B" remains intact, and the sentence is more natural.

<...>

**Demonstrations:**

Input:

<Formal name><Formal statement><Informal name>< Informal statement><Head statements><Docstring><Dependent statements><Tactic explanation><Proof state>

Output:

<Informal proof steps><Whole informal proof>

Figure 8: Prompt for informalizing Mathlib4 proof

