# OpenReview forum: "Herald: A Natural Language Annotated Lean 4 Dataset"
_ICLR.cc/2025/Conference — ICLR 2025 Poster_

### Official Review · Reviewer_4Ljo · 2024-10-30

**Soundness:** 3
**Presentation:** 2
**Contribution:** 3
**Rating:** 6
**Confidence:** 4

**Summary:**

The authors approach the problem of autoformalization, a heavily data-scarce domain, by automatically generating a dataset of natural language-formal language pairs by auto-informalizing mathlib. This is done with LLMs which are fed important context such as dependent theorems, docstrings, and similarly-named theorems (which are often highly related) together with a retrieval-augmented system to improve translation quality.

Furthermore, they augment proof data to acquire similar data for NL-FL proof pairs in two ways. First they sample intermediate proofs, and second they paraphrase informal statements.

After this data is compiled, a formalization LLM is fine-tuned on the resulting dataset, and a pretrained theorem prover is used to generate the proofs, and the authors show this data is useful for improving autoformalization performance on their chosen benchmarks.

**Strengths:**

The proposed dataset has clear use to the community as a large dataset of natural language/formal language pairs for autoformalization system training. Additionally, the Herald translator is an improvement over previous methods, achieving very high performance on translating miniF2F problems, and improved performance on more difficult college-level datasets. Overall, I believe the work is well-supported.

**Weaknesses:**

- The authors use a pass@128 metric for successful autoformalization. This seems like a highly lenient metric, and I'm curious to know if there is prior work which uses this metric, or if the authors chose this for a particular reason (I don't recall any previous use of this in autoformalization, but I may be wrong). If high performance is contingent on so many trials, I believe the compute efficiency is a significant weakness of the approach.
- Similarly, are the baseline approaches (Theorem Llama, Llama-3 instruct, etc.) in Table 2 also measured using Pass@128?
- It is unclear whether Table 2 is autoformalization performance is only theorem statements, or both theorems and proofs. MiniF2F contains some formalized proofs, but isn't complete, and I am unfamiliar with the other two. If Table 2 is only theorem statements, are there results showing the proof-translation performance?
- There are no examples of generated pairs from the Herald dataset which makes it difficult to assess the data quality. Examples from the two augmentation would be especially insightful. I see outputs from the Herald translator in the Appendix, which look high quality.
- Also, a comparison of the Herald Translator trained on various datasets is important to judge the quality of the Herald dataset. For example, trained on MMA, LeanWorkbook, etc. and evaluated on the same benchmarks as the Herald Translator was.

**Questions:**

- See "Weaknesses" section above for some potential places to improve clarity of paper.
- Is there a reason you didn't evaluate on ProofNet (there are available Lean 4 versions of the dataset)? It is a useful benchmark for more difficult undergraduate problems, which seems to be in-line with your other choices.
- What was your selection process for the Stacks project? From the outside, it looks like a cherry picked example, which I think is ok but it should be made clear.
- I am unclear as to what the aligned proof data from the Herald was used for in the training of the model. It seems like all the reported experiments are just for theorem statement formalization. Can you clarify?

Overall, I believe this can be a solid paper but needs some clarifications. Assuming the clarifications don't significantly harm the paper and the experiments and results are sound I think my rating(s) can be improved.

---

> ### Author Response · Authors · 2024-11-27
> **Response to Reviewer 4Ljo (1/3)**
>
> **R W1 & W2: Pass@128 metric**
>
> Thank you for raising this important point. First, we would like to emphasize that **all baselines reported in Table 2, including TheoremLlama, InternLM2-Math-Plus-7B, and Llama3-Instruct, are evaluated under the same Pass@128 setting**. This consistent evaluation is critical for ensuring a fair and direct comparison of the performance of our method relative to existing approaches.
> Regarding the use of Pass@128, this metric is consistent with established practices in the auto-formalization literature:
> - In [1], the authors translate problems from the Compfiles dataset 100 times, leveraging diverse sampling to improve success rates.
> - In [2], Pass@100 is explicitly employed as the evaluation metric.
> - In [3] and [4], Pass@128 or higher budgets are used to maximize the potential of large language models in generating diverse outputs, especially for harder problems.
>
> We adopted Pass@128 to align with these established practices, as higher sampling budgets are particularly important for tackling challenging auto-formalization tasks where generating a diverse range of candidate solutions is essential.
> Additionally, we conducted an ablation study where we evaluated the performance of our Herald Translator under Pass@k settings (k = 16, 32, 64, 128). The results are summarized in the table below. As shown, Herald Translator consistently outperforms the best baseline (InternLM2-Math-Plus-7B) across all Pass@k settings. Notably, Herald Translator at Pass@16 already surpasses the baseline at Pass@128, demonstrating the robustness and efficiency of our method. (Note that we chose a checkpoint of Herald Translator better than the one selected in the previous version of this paper, which is already updated in the new version.)
>
> | Model                  | Pass | ProofNet-valid | ProofNet-test | MiniF2F-test | Extract-Theorem |
> | ---------------------- | ---- | -------------- | ------------- | ------------ | --------------- |
> | Herald Translator      | 16   | 60.5           | 66.7          | 84.8         | 9.5             |
> | Herald Translator      | 32   | 68.1           | 76.9          | 91.8         | 14.0            |
> | Herald Translator      | 64   | 76.8           | 83.9          | 95.5         | 18.5            |
> | Herald Translator      | 128  | 81.6           | 88.7          | 96.7         | 23.5            |
> | InternLM2-Math-Plus-7B | 16   | 13.5           | 15.1          | 52.0         | 2.5             |
> | InternLM2-Math-Plus-7B | 32   | 20.0           | 25.3          | 61.1         | 3.0             |
> | InternLM2-Math-Plus-7B | 64   | 33.0           | 36.0          | 67.5         | 5.0             |
> | InternLM2-Math-Plus-7B | 128  | 38.9           | 45.2          | 73.0         | 7.5             |
>
> We present the results for ProofNet-valid here because we have included a section for the case study on ProofNet-valid results in the appendix. As shown, the results of ProofNet-test are similar to those of ProofNet-valid. We hope this explanation provides sufficient context for our choice of metric and highlights the robustness of Herald Translator across different sampling budgets.
>
> [1] Ying, Huaiyuan, et al. "Lean Workbook: A large-scale Lean problem set formalized from natural language math problems." NeurIPS, 2024
>
> [2] Jiang, Albert Qiaochu, et al. "Draft, Sketch, and Prove: Guiding Formal Theorem Provers with Informal Proofs." ICLR, 2023
>
> [3] Wang, Ruida, et al. "Theoremllama: Transforming general-purpose LLMs into lean4 experts." _arXiv preprint arXiv:2407.03203_ (2024).
>
> [4] Xin, Huajian, et al. "DeepSeek-Prover-V1. 5: Harnessing Proof Assistant Feedback for Reinforcement Learning and Monte-Carlo Tree Search." _arXiv_ _preprint_ _arXiv:2408.08152_ (2024).

---

> ### Author Response · Authors · 2024-11-27
> **Response to Reviewer 4Ljo (2/3)**
>
> **R W3: auto-formalization performance for proofs**
>
> To clarify, Table 2 in Section 4.1 reports the results of our statement formalization model, focusing solely on theorem statements rather than proofs.
>
> Unfortunately, we did not include quantitative results for proof formalization due to limitations in the dataset. The proof informalization-to-formalization pipeline proposed in our paper is specifically designed for tactic-based proofs, which account for only approximately one-fourth of the theorems in Mathlib4. By "tactic-based proof," we refer to proofs whose top-layer logic relies on tactics (i.e., proofs that begin with `:= by`). While Mathlib4 contains merely around 45k such proofs, this number is insufficient to fine-tune a proof auto-formalization model capable of achieving strong performance on theorem-proving datasets like MiniF2F and ProofNet.
>
> This limitation arises from an intrinsic characteristic of Mathlib4, where contributors often prefer to decompose theorems into simpler lemmas and write termwise proofs for these simpler lemmas to optimize compilation efficiency.
>
> That said, we emphasize that the proposed informalization pipeline for proofs produces high-quality informal-formal stepwise annotations. This is demonstrated through case studies in Appendix E.2, where our method shows notable improvements in translation quality, particularly in terms of mathematical validity and alignment. By providing large language models (LLMs) with sufficient context, our pipeline greatly enhances the quality of the generated proofs.
>
> While this paper primarily focuses on statement informalization, augmentation, and formalization, we believe our proof informalization pipeline is a valuable tool that can be readily applied to any Lean code corpus to generate informal annotations for proofs. Future work will aim to further develop and extend this pipeline, enabling more comprehensive informalization and augmentation for formal proofs.
>
> **R W4: Examples for Herald dataset**
>
> We have added real data pairs in Appendix E and included additional examples for each augmentation method with analysis. Here are some examples copied from the appendix.
>
> A. Original statements in mathlib
>
> Example 1
>
> `theorem CongruenceSubgroup.Gamma_zero_bot : CongruenceSubgroup.Gamma 0`
>
> The full level 0 congruence subgroup of SL(2, Z), denoted by Γ(0), is equal to the trivial
>
> subgroup, i.e., Γ(0) = {1}.
>
> Example 2
>
> ```
> theorem dite_eq_or_eq {α : Sort u_1} (P : Prop) [inst : Decidable P] {A : P → α} {B : ¬P → α} :
>     (∃ h, dite P A B = A h) ∨ ∃ h, dite P A B = B h
> ```
>
> Dependent If-Then-Else Equals One of the Branches: For any type $\alpha$, any proposition $P$, and any decidable instance of $P$, if $A : P \to \alpha$ and $B : \neg P \to \alpha$ are functions, then the dependent if-then-else construct `dite P A B` is equal to either $A(h)$ for some proof of $P$, or $B(h)$ for some proof of $\neg P$. In other words, the value of `dite P A B` is either the value of $A$ when $P$ is true, or the value of $B$ when $P$ is false.
>
> B. Informal augmented statements
>
> ```
> theorem sum_not_convergent'_tac : Tendsto (fun n => ∑ i in Finset.range n, 1/i) atTop atTop := by sorry
> ```
>
> Let $n$ be positive natural number. The infinite sum of the sequence $\frac{1}{n}$ is not convergent.
>
> This statement is augmented to:
>
> 1. (Logical Equivalence Rewriting) The infinite sum of the sequence $\frac{1}{n}$ is divergent, where $n$ is a positive natural number.
>
> 2. (Abstract Concept Substitution) Let $n \in \mathbb{N}^+$. The harmonic series diverges.
>
> 3. (Omission of Implicit Condition) The sequence `1/n` does not have a convergent infinite sum.
>
> 4. (Multi-linguistic Translation) Soit $n$ un nombre naturel positif. La somme infinie de la suite $\frac{1}{n}$ n'est pas convergente.
>
>
> C. Tactic augmentation
>
> Given the proof of Cantor's theorem
>
> ```
> theorem cantor (a : Cardinal.{u}) : a < 2 ^ a := by
>   induction' a using Cardinal.inductionOn with α
>   rw [← Cardinal.mk_set]
>   refine ⟨⟨⟨singleton, fun a b => singleton_eq_singleton_iff.1⟩⟩, ?_⟩
>   rintro ⟨⟨f, hf⟩⟩
>   exact cantor_injective f hf
> ```
>
> We extract the proof states from each line and get new formal-informal statement pairs.
>
> Example 1:
>
> `lemma cantor_aug_2 : Cardinal.mk α < Cardinal.mk (Set α) := sorry`
>
> For any set $\alpha$, show that the cardinality of $\alpha$ is strictly less than the cardinality of the power set of $\alpha$, i.e., $|\alpha| < |2^{\alpha}|$.
>
> Example 2:
>
> `lemma cantor_aug_4 (f : Set α → α) (hf : Injective f) : False := sorry`
>
> For any function $f$ from the power set of $\alpha$ to $\alpha$, and given that $f$ is injective, show that the statement leads to a contradiction, i.e., it is false.

---

> ### Author Response · Authors · 2024-11-27
> **Response to Reviewer 4Ljo (3/3)**
>
> **R W5: Comparision to training Herald translator on a different dataset (e.g. MMA, LeanWorkbook)**
>
> Thank you very much for this valuable suggestion. We agree that such an experiment would provide strong evidence to further demonstrate the effectiveness of our work. However, due to time constraints, we are unable to include these results before the discussion deadline, as performing full-parameter supervised fine-tuning (SFT) on the Herald dataset requires approximately 240 A800 GPU hours. We appreciate your understanding and your insightful feedback, which will undoubtedly help strengthen our work.
>
> **R Q1:**
>
> See previous sections.
>
> **R Q2: Evaluation on ProofNet**
>
> Thank you for highlighting this important benchmark. We have now included the results for ProofNet in our main results, which are presented in the table in Section R W1 & W2.
>
> As shown in the updated table, our translator demonstrates similarly outstanding performance on this undergraduate-level problem set. To further support these findings, we have added a section including case studies on the formalization results of ProofNet in Appendix D.
>
> Our translator achieves much higher performance on ProofNet compared to Extract Theorem. We attribute this difference to the well-established Lean infrastructure available in Mathlib4 for ProofNet statements, which facilitates clean and consistent formalization. This robust infrastructure likely also explains why these statements are suitable for use as a formalization benchmark. In contrast, the Extract Theorem dataset consists of real-world undergraduate-level problems, many of which lack the necessary infrastructure in Lean. This gap makes these problems more challenging for both human formalization and auto-formalization.
>
> **R Q3: Selection process for the Stacks project**
>
> We selected the chapter based on the following principles:
>
> - Pre-existing Formalized Definitions: We ensured that the definitions referenced without explanation in the selected section were already formalized in Mathlib, as this minimizes additional assumptions and ensures a clearer evaluation of our method.
>
> - Avoidance of Complex TFAE Theorems: Sections containing large "The Following Are Equivalent" (TFAE) theorems with 10+ equivalent conditions were avoided to focus on sections that allowed for a more straightforward demonstration of our approach.
>
>
> We invited mathematical experts to compile a list of sections from the Stacks Project that aligned with these criteria. The example we included in the appendix is the first entry from this list.
>
> **R Q4: About aligned proof data from the Herald**
>
> See Section R W3. The proof data is not used in the quantitative experiment and thus only exhibited in the case study section.

---

> > ### Comment · Reviewer_4Ljo · 2024-11-30
> > **Response to rebuttal**
> >
> > Thank you authors for carefully addressing my concerns with your work, and for pointing out the examples I missed in the appendix.
> >
> > After reviewing the updated information, most of my concerns are satisfied and I have updated my overall review from a 5 to 6. My remaining concerns are due to the clarity of statement vs proof autoformalization (which the authors have made clear in the above comments, but should still be amended in the paper in my view). Additionally, I believe including the Pass@K results above would make an even stronger argument for the effectiveness of your method.

---

### Official Review · Reviewer_BoSn · 2024-11-01

**Soundness:** 3
**Presentation:** 3
**Contribution:** 3
**Rating:** 6
**Confidence:** 4

**Summary:**

- This paper develops a novel parallel natural language-to-formal language data synthesis method and extends it into a practical method for automatic formalization of literature.
- The method demonstrates strong generalization ability with significant performance gains.
- This tool will be useful for future data synthesis in the community.

**Strengths:**

- The paper extends the boundaries of autoformalization using a Retrieval-Augmented Generation (RAG) based method.
- The paper develops an innovative, practical tool that helps extend its reach to users beyond the community.
- The paper's method is simple, and the writing is well-done.

**Weaknesses:**

- I believe that a contextual learning approach was used in the translation process. There should be a detailed investigation into how different retrieval methods affect it, as they directly impact the accuracy of the model's output.
- Why not compare it with other autoformalization methods, such as [1,2].



**Reference**
[1] FVEL: Interactive Formal Verification Environment with Large Language Models via Theorem Proving.
[2] FormalAlign: Automated Alignment Evaluation for Autoformalization.

**Questions:**

- Please include some additional baselines.
- Ablation studies are missing; please include them.

---

> ### Author Response · Authors · 2024-11-27
> **Response to Reviewer BoSn (1/2)**
>
> **R W1: Clarification on retrieval methods used in translation**
>
> As it is unclear whether your question pertains to the formal → informal translation process in dataset generation or the informal → formal process in auto-formalization, we will address both below.
>
> - Formal → Informal Translation (Herald Dataset Creation): During the creation of the Herald dataset, we employed a contextual learning approach, which is detailed in Section 2 of our paper. Conducting a full ablation study to analyze the impact of different components of our RAG system is not practical due to the size of the input data, which amounts to approximately 2.2 billion tokens. However, we have carefully selected examples to demonstrate the effectiveness of our retrieval system, as shown in Appendix E.
>
>     - In Appendix E.1.1, the example involving the acronym "dite" highlights the necessity of hierarchical dependency retrieval in the informalization process. Without the retrieved contextual information, the LLM would be forced to guess the meaning of "dite", leading to inaccurate results. This example directly inspired the introduction of our RAG system.
>
>     - Similarly, Appendix E.1.4 illustrates the impact of human-written head statements in the context of category theory. These detailed head statements enable the LLM to correctly interpret the complex setup of the commutative diagram, further demonstrating the value of our retrieval system.
>
> - Informal → Formal Translation (Herald Translator Evaluation): In the experiments for statement formalization, the input consisted solely of the LaTeX code representing the theorem statements, and **no** in-context learning or retrieval-based methods were employed. Additionally, during the training of the Herald Translator, no retrieval components were included in the training data, ensuring the model's output relies purely on the given theorem statements without external context.
>
>
> We hope this clarification addresses your concerns and provides a detailed explanation of our approach to retrieval methods in both processes.
>
> **R W2: Comparision to other auto-formalization methods**
>
> Thank you for suggesting these potential methods for comparison to further validate the value of our approach. We have carefully investigated the papers you mentioned.
>
> We found that the auto-formalization methods in [1] were implemented using Isabelle, which relies on different infrastructures and mathematical libraries compared to Lean. This difference makes a direct comparison not entirely fair. As for [2], its arXiv release is very recent (14 October 2024) and focuses on detecting misalignments between informal and formal statements, rather than performing auto-formalization tasks.
>
> To the best of our knowledge, TheoremLlama is the most closely related work addressing both the informalization of Mathlib4 and auto-formalization. We have compared our results against it both quantitatively (Section 4.1.3) and qualitatively (Appendix E).
>
> **R Q1: Additional baselines:**
>
> Thank you for this suggestion! Beyond our original results, we evaluated the performance of gpt-4o-2024-08-06 on the ProofNet-valid and MiniF2F-test datasets, as summarized below:
>
> | Model             | Pass | ProofNet-valid | MiniF2F-test |
> | ----------------- | ---- | -------------- | ------------ |
> | Herald Translator | 128  | 81.6           | 96.7         |
> | gpt-4o-2024-08-06 | 128  | 0.00           | 1.19         |
>
> The performance of GPT-4o on Lean 4 formalization tasks is extremely poor. Despite being clearly instructed to use Lean 4 for formalization, the model frequently mixes Lean 4 and Lean 3 syntax in its outputs. This inconsistency results in outputs that are often incompatible with the Lean 4 compiler, as highlighted in Appendix A of [1].
>
> This limitation underscores a key challenge in using proprietary large language models (LLMs) with prompt-based techniques for formalization tasks. It is one of the main reasons researchers develop and fine-tune their own auto-formalizers instead of relying on these general-purpose LLMs.
>
> [1] Wang, Ruida, et al. "Theoremllama: Transforming general-purpose llms into lean4 experts." _arXiv_ _preprint_ _arXiv:2407.03203_ (2024).

---

> ### Author Response · Authors · 2024-11-27
> **Response to Reviewer BoSn (2/2)**
>
> **R Q2: Ablation study**
>
> Thank you for highlighting this point. We conducted an ablation study on the two augmentation methods we proposed and evaluated performance under different sampling budgets. The results are as follows:
>
> | Dataset ratio/Model    | Dataset size | Pass | ProofNet-valid | MiniF2F-test | Extract-Theorem |
> | ---------------------- | ------------ | ---- | -------------- | ------------ | --------------- |
> | 1:2:1                  | 1160k         | 16   | 60.5           | 84.8         | 9.5             |
> | 1:2:1                  | 1160k         | 32   | 68.1           | 91.8         | 14.0            |
> | 1:2:1                  | 1160k         | 64   | 76.8           | 95.5         | 18.5            |
> | 1:2:1                  | 1160k         | 128  | 81.6           | 96.7         | 23.5            |
> | 1:2:1                  | 580k         | 16   | 60.0           | 83.2         | 7.5             |
> | 1:2:1                  | 580k         | 32   | 67.6           | 88.5         | 10.5            |
> | 1:2:1                  | 580k         | 64   | 75.7           | 90.6         | 14.5            |
> | 1:2:1                  | 580k         | 128  | 78.6           | 93.2         | 20.5            |
> | 1:1:1                  | 435k         | 16   | 42.5           | 82.8         | 10.0            |
> | 1:1:1                  | 435k         | 32   | 51.1           | 88.1         | 12.0            |
> | 1:1:1                  | 435k         | 64   | 61.3           | 90.2         | 14.0            |
> | 1:1:1                  | 435k         | 128  | 68.3           | 91.4         | 17.5            |
> | 1:2:0                  | 435k         | 128  | 74.7           | 95.1         | 19.0            |
> | 1:0:0                  | 145k         | 128  | 50.0           | 86.8         | 15.0            |
> | InternLM2-Math-Plus-7B | N/A          | 16   | 13.5           | 52.0         | 2.5             |
> | InternLM2-Math-Plus-7B | N/A          | 32   | 20.0           | 61.1         | 3.0             |
> | InternLM2-Math-Plus-7B | N/A          | 64   | 33.0           | 67.5         | 5.0             |
> | InternLM2-Math-Plus-7B | N/A          | 128  | 38.9           | 73.0         | 7.5             |
>
>
> In the table, the ratio represents `original:tac_aug:inf_aug`, where:
>
> - original denotes the informal-formal pairs from the translated Mathlib4 dataset,
>
> - tac_aug refers to the tactic-based augmented statements, and
>
> - inf_aug corresponds to the informal augmented statements.
>
>
> All groups share the same "original" set, and we ablate the augmented datasets to assess the effectiveness of our methods. As a result, each group contains a different size of training data. The base model is trained on each dataset separately, and the checkpoint with the best performance is selected for each group. The model with the 1:2:1 ratio and 580k dataset size is trained on NL→FL dataset, and the model with the 1:2:1 ratio and 1160k dataset size is trained on NL→FL and FL→NL datasets. Note that during the ablation test, we discovered a checkpoint with the 1:2:1 ratio (the main experimental setting in our paper) that outperforms the one originally selected in the paper. We have already updated the results in the new version of the paper using this checkpoint.
>
> As shown in the table, our augmentation methods effectively enhance model performance, particularly on more challenging datasets such as ProofNet. The model trained solely on the original informal-formal pairs (1:0:0) achieves a respectable accuracy of 50.0\% on the ProofNet-valid dataset, 86.8\% on the MiniF2F-test dataset, and 15.0\% on the Extract-Theorem dataset, which underscores the quality of the informalized Mathlib4 dataset.
>
> The introduction of augmented datasets significantly boosts model performance across all datasets. The model trained with a 1:1:1 ratio shows a notable improvement, achieving 68.3\% on ProofNet-valid, 91.4\% on MiniF2F-test, and 17.5\% on Extract-Theorem. The 1:2:0 ratio model, which includes more tactic-based augmented statements, further enhances performance, reaching 74.7\% on ProofNet-valid, 95.1\% on MiniF2F-test, and 19.0\% on Extract-Theorem.
> The model trained with 1:2:1 ratio on both NL→FL and FL→NL datasets outperforms the model trained with 1:2:1 ratio on only NL→FL datasets, with accuracies of 81.6\% on ProofNet-valid, 96.7\% on MiniF2F-test, and 23.5\% on Extract-Theorem. This result indicates that increasing the size of the training set, while maintaining a balanced augmentation ratio, leads to substantial improvements in model performance, especially on more challenging datasets.
>
> We have integrated the ablation study results, along with the updated main results for the Herald Translator, into the new version of our paper.

---

### Official Review · Reviewer_6Lbv · 2024-11-03

**Soundness:** 2
**Presentation:** 3
**Contribution:** 3
**Rating:** 8
**Confidence:** 4

**Summary:**

This paper describes an autoinformalization (formal->natural translation) pipeline for Lean 4 statements and proofs. The authors also contribute Herald, an LM fine-tuned for autoformalization on a dataset of formal-informal statement pairs generated by applying the proposed NL augmentation pipeline to mathlib4, the Lean 4 math library.

The authors' pipeline retrieves relevant definitions and documentation in order to ensure uninformative formal names can be translated appropriately. When translating statements as part of proof informalization, their method analyzes proof structure in order to stratify statements (i.e. sort them topologically) ensuring that natural versions of a statement's logical dependencies are available when translating it.

The authors include qualitative comparisons between their method and prior art (Open Bootstrapped Theorems/OBT) detailing both successes and failure cases.

The authors evaluate their fine-tuned autoformalization model by computing the proportion of statements in a test set for which any of a set of 128 samples drawn from the formalization model pass two checks: syntactic correctness according to the Lean REPL, and backtranslation with a fixed informalization LM followed by LM-based NLI between the original and backtranslated NL. This evaluation is conducted on three test sets: miniF2F, a custom dataset extracted from textbooks via OCR, and another one scraped and filtered from internet math resources. The authors compare their fine-tuned model to three baseline models: TheoremLlama, InternLM2-Math-Plus-7b, and Llama 3 Instruct. According to the authors' chosen metric, their model outperforms the other considered models substantially across all three sets.

**Strengths:**

- The proposed augmentation pipeline is well-designed. In particular, it's a great idea to use the results of structural analysis to order statement translation within proofs to allow informalization to condition on both formal and natural language versions of a statement's logical context. Herald definitely seems to me to be a step forward in terms of the quality of augmented examples.

- While many of the comparisons this paper makes between Herald and prior work are qualitative, I found the detailed example-based analysis in the appendix to be convincing support for these claims. I greatly appreciate the authors' candor in highlighting Herald's shortcomings in addition to positive differences between their method and OBT.

**Weaknesses:**

- It's hard for me to get a sense of the significance of the fine-tuning results, given that both the chosen metric and two of the three datasets used for evaluation are constructed by the authors. It would be helpful to see some representative examples of the two custom test sets, in addition to samples from the model that pass and fail each validation step.

- I also feel that it's essential to audit the LM NLI check and report its correlation with human expert judgements of semantic equivalence for original + back-translated statements from the test domains (it's fine if this is a reasonably small sample size). Without this information, the valid% metric seems shaky.

**Questions:**

1. It seems like one of the main issues remaining in the Herald natural augmentations is the presence/copying of formal names. From the text and the prompts it sounds like you mainly attempted to combat this using instructions, but that had limited power. Did you try rejection sampling (sample multiple times and reject translations containing Lean-specific names)?

2. What is the specific provenance of the web-scraped test set (College CoT)?

---

> ### Author Response · Authors · 2024-11-27
> **Response to Reviewer 6Lbv (1/3)**
>
> **R W1: Examples of the Custom Test Sets**
>
> Following your suggestion, we have included examples of the **Extract Theorem** and **College** **CoT** datasets in Appendix G. Additionally, we provide some examples and a summary here. Note that the full dataset is available in the supplementary material of the paper.
>
> **Extract Theorem**
>
> **Example 1:** Lemma 1.7 Suppose that $p \geq 3$ is a prime. Then all non-normal subgroups of a finite $p$-group $G$ are conjugate if and only if $$ G = \langle u, v \mid u^{p^n} = 1, v^p = 1, v^{-1}uv = u^{1 + p^{n - 1}}, n \geq 2 \rangle. $$
>
> **Example 2:** Assume that a gambler making fair unit bets on coin flips will abandon the game when her fortune falls to 0 or rises to $n$. Let $X_t$ be the gambler’s fortune at time $t$, and let $\tau$ be the time required to be absorbed at one of $0$ or $n$. Assume that $X_0 = k$, where $0 \leq k \leq n$. Then:
>
>
> $$ \mathbb{P}_k  ( X\_{\tau} = n ) = \frac{k}{n} \tag{2.1} $$
>
>
> $$ \mathbb{E}_k(\tau) = k(n - k) \tag{2.2} $$
>
> **College CoT**
>
> **Example 1:** Theorem 31 (Nash & Tognoli) Any smooth compact manifold is diffeomorphic to a smooth real algebraic submanifold of some $\mathbb{E}^n$, i.e., to the set of solutions of a system of polynomial equations.
>
> **Example 2:** Theorem 9.3.1 (Hahn-Kolmogorov Extension Theorem — Uniqueness of Extension of a $\sigma$-Finite Measure) Let $\mathcal{A}$ be an algebra of subsets of a non-empty set $X$, and let $m : \mathcal{A} \to [0, \infty)$ be a $\sigma$-finite measure on $\mathcal{A}$. There exists a unique extension of $m$ to a measure $m'$ defined on the $\sigma$-algebra $\mathcal{S}(\mathcal{A})$.
>
> **Summary**
> - The **Extract Theorem** dataset includes a broad spectrum of advanced undergraduate-level material, including but not limited to algebra, analysis, geometry, and probability theory.
> - The **College CoT** dataset features more advanced graduate-level mathematical materials (e.g., algebraic geometry, measure theory) compared to the Extract Theorem dataset.
>
>
> **Case Study of Validation Pipeline**
>
> For the remaining part of this question, we present representative examples from the model that pass and fail each validation step. We have added a case study section in Appendix D.2 for detailed inspection. Below, we copy some examples for reference:
>
> **Example 1 (Accepted by Validations):**
>
> Suppose $T \in \mathcal{L}(V)$ is such that every vector in $V$ is an eigenvector of $T$. Prove that $T$ is a scalar multiple of the identity operator.
>
> ```
> theorem isEigenvector_spec (V : Type*) [AddCommGroup V] [Field K] [Module K V] (T : V →ₗ[K] V) (h : ∀ v : V, ∃ k : K, k • v = T v) : ∃ k : K, ∀ v : V, k • v = T v := by sorry
> ```
>
> **Example 2 (Failing Lean Validation):** Show that a closed subspace of a normal space is normal.
>
> ```
> theorem isNormal_subspace_of_normalSpace {t : TopologicalSpace X} [NormalSpace X] (S : Set X) (hS : IsClosed S) : IsNormal S := sorry
> ```
>
> The predicate `IsNormal` is not defined in mathlib.
>
> **Example 3 (Failing NLI Validation):**
>
> Show that a closed subspace of a normal space is normal.
>
> ```
> theorem munkres_32_1 {X : Type*} [TopologicalSpace X] {S : Set X} [NormalSpace X] (T : Set S) (hT : IsClosed T) : NormalSpace S := sorry
> ```
>
> Back translation
>
> Let $X$ be a topological space; let $S$ be a subspace of $X$. Suppose that $X$ is normal. Show that $S$ is normal.
>
> The NLI check identifies that the condition $S$ is closed is missing, thereby rejecting the translation.

---

> ### Author Response · Authors · 2024-11-27
> **Response to Reviewer 6Lbv (2/3)**
>
> **R W2: Human expert examination of the LM** **NLI** **check**
>
> Following your recommendation, we conducted a human expert examination of the successful results for the LM NLI check, constrained to the ProofNet validation set (185 statements in total). We reviewed every statement that passed both the Lean compiler and the LM NLI check. Below is a table summarizing the results. For comparison, we included InternLM2-Math-Plus-7B, the baseline model with the best performance in our experiments.
>
>
> | Model                  | Correct | Minor error | Major error |
> | ---------------------- | ------- | ----------- | ----------- |
> | InternLM2-Math-Plus-7B | 42      | 12          | 18          |
> | Herald Translator      | 101     | 24          | 26          |
>
> The auto-formalization outcomes are classified into three distinct categories by human experts: **correct translations**, **minor errors**, and **major errors**:
> - **Correct Translation**: The formalization must precisely capture the mathematical meaning of the natural language statement. If the original statement is ambiguous, the formalized statement may adopt any mathematically valid interpretation.
> - **Minor Error**: The formalized result deviates slightly from the original statement but can be corrected by modifying a single formal definition or restructuring a single logical relation within the hypothesis or conclusion.
> - **Major Error**: Any other deviations that require more extensive corrections fall into this category.
>
> A detailed inspection of Herald Translator’s outputs across all three categories has been added in Appendix D.
>
> It is important to note that even the strictest LLM-based NLI checking introduces false positives. LLMs are less capable than humans at detecting subtleties in formalized statements, such as filling back opened namespaces and disambiguating overloading notations and they may fail to decide whether two NL statements align due to a lack of mathematical understanding and the various possibilities of stating mathematics in neutral language. Consequently, formalized results passing the NLI check may fail to reflect the mathematical meaning of the original natural language statement (i.e., false positives). Therefore, NLI checking is not as reliable as human expert evaluation.
>
> For our application of the NLI checking pipeline, the most important goal is to create a fair performance test between different models. As we see in the table above, both the baseline model and our model achieve similar results in terms of the proportion of false-positive cases.
>
> However, while NLI checking is not perfectly accurate for evaluating informal-to-formal alignment, it remains a common practice in the field (e.g., see [1]) and one of the few practical approaches for large-scale automated evaluation at this time. It is not practical to apply human evaluation to a dataset with more than 1k formalization results. It is also worth noting that NLI check may introduce false negatives. While it is impractical to verify all negative cases, we include the following example to illustrate:
>
> **Informal statement**
>
> Show that if $\prod X_\alpha$ is regular, then so is $X_\alpha$. Assume that each $X_\alpha$ is nonempty.
>
> **Generated formal statement**
>
> ```
>
> theorem regularSpace_of_product_regularSpace {ι : Type*} {X : ι → Type*} [∀ i, TopologicalSpace (X i)] [∀ i, Nonempty (X i)] (h : RegularSpace (Π i, X i)) : ∀ i, RegularSpace (X i) := sorry
>
> ```
>
> **Back translation**
>
> Let $X_i$ be a regular space for each $i \in I$. Prove that the product space $\prod_{i \in I} X_i$ is regular.
>
> In this example, the Herald Translator accurately formalized the statement: the product space being regular implies that each component is regular. However, the back translation reversed the direction of the implication, resulting in a failed NLI check.
>
> As shown in the table, other baseline models also suffer from false positives, with comparable false positive rates, making the comparison fair. While NLI checking is not perfectly accurate for evaluating informal-to-formal alignment, it remains a common practice in the field ([1]) and one of the few practical approaches for large-scale automated evaluation at this time. We are excited by recent advancements, such as [2], and plan to incorporate these methods into future work.
>
> [1] Ying, Huaiyuan, et al. "Lean Workbook: A large-scale Lean problem set formalized from natural language math problems." NeurIPS, 2024
>
> [2] Lu, Jianqiao, et al. "FormalAlign: Automated Alignment Evaluation for Autoformalization." _arXiv_ _preprint_ _arXiv:2410.10135_ (2024).

---

> ### Author Response · Authors · 2024-11-27
> **Response to Reviewer 6Lbv (3/3)**
>
> **R Q1: Additional methods for improving informalization**
>
> Thank you for your valuable advice. We explored two potential methods to enhance the quality of our results: (1) sampling to select the best result from four attempts and (2) introducing an LLM critic based on previous LLM outputs. However, a small-scale case study revealed that the language style of the generated content remained largely consistent across different attempts and showed minimal improvement after applying the critic.
>
> Additionally, implementing these methods would significantly increase computational cost—at least quadrupling or doubling it—while offering only limited improvements in quality. Given this trade-off, we decided not to include these methods in our final approach.
>
> Moreover, since the language style of the generated content remains unchanged, rejection sampling may require a large number of retries, further increasing computational costs. In future work, we aim to explore the balance between informalization cost and data quality to refine our methods.
>
> **R Q2: Provenance for the name "College CoT"**
>
> Thank you for raising this question. The term "College" reflects the difficulty level of the mathematical materials we collected, while "CoT" stands for "Chain of Thought," referring to the method we used to prompt the LLM to structure the material into the desired isolated theorem-proof format. However, in this paper, we only utilize the theorem statements. It is important to note that the use of "CoT" here is unrelated to the experiments discussed in the paper. To avoid potential confusion, we will revise this name for better clarity.

---

> ### Comment · Reviewer_6Lbv · 2024-12-02
>
> Thanks so much for your detailed responses!

---

### Official Review · Reviewer_5xV4 · 2024-11-11

**Soundness:** 3
**Presentation:** 3
**Contribution:** 3
**Rating:** 8
**Confidence:** 3

**Summary:**

This paper proposes an informalization pipeline based on static analysis of lean, which can be used to create parallel datasets of natural language proofs and formal language proofs.
They used their pipeline to create a natural language version of Mathlib4, and trained a transformer that formalizes natural language statements to lean.
The trained model, Herald Translator, is much better than existing models on auto-formalization.

**Strengths:**

1. The authors provide a clear rationale for their methods and promise to open-source the whole pipeline for data generation. This is helpful for future research in AI-assisted math.
2. The evaluation results look impressive, and they are beyond standard benchmarks. The authors demonstrated Herald translator's effectiveness on a real-world formalization project.

**Weaknesses:**

1. Lack of ablation studies. It's not clear to me which parts of the data generation/augmentation are helpful. More ablation experiments would help, especially for the multilinguistic augmentation. I'm really curious if that would result in NL statements in different languages.
2. Does DeepSeek-Prover-1.5-Instruct perform well on formalization? If so, it would be interesting to see how herald translator compares with it.
3. Contamination: Could the data augmentation/curation process introduce contamination issues such as putting parts of miniF2F into the training set?

**Questions:**

See weaknesses.

---

> ### Author Response · Authors · 2024-11-27
> **Response to Reviewer 5xV4 (1/2)**
>
> **R W1: Ablation study**
>
> Thank you for highlighting this point. We conducted an ablation study on the two augmentation methods we proposed and evaluated performance under different sampling budgets. The results are as follows:
>
> | Dataset ratio/Model    | Dataset size | Pass | ProofNet-valid | MiniF2F-test | Extract-Theorem |
> | ---------------------- | ------------ | ---- | -------------- | ------------ | --------------- |
> | 1:2:1                  | 1160k         | 16   | 60.5           | 84.8         | 9.5             |
> | 1:2:1                  | 1160k         | 32   | 68.1           | 91.8         | 14.0            |
> | 1:2:1                  | 1160k         | 64   | 76.8           | 95.5         | 18.5            |
> | 1:2:1                  | 1160k         | 128  | 81.6           | 96.7         | 23.5            |
> | 1:2:1                  | 580k         | 16   | 60.0           | 83.2         | 7.5             |
> | 1:2:1                  | 580k         | 32   | 67.6           | 88.5         | 10.5            |
> | 1:2:1                  | 580k         | 64   | 75.7           | 90.6         | 14.5            |
> | 1:2:1                  | 580k         | 128  | 78.6           | 93.2         | 20.5            |
> | 1:1:1                  | 435k         | 16   | 42.5           | 82.8         | 10.0            |
> | 1:1:1                  | 435k         | 32   | 51.1           | 88.1         | 12.0            |
> | 1:1:1                  | 435k         | 64   | 61.3           | 90.2         | 14.0            |
> | 1:1:1                  | 435k         | 128  | 68.3           | 91.4         | 17.5            |
> | 1:2:0                  | 435k         | 128  | 74.7           | 95.1         | 19.0            |
> | 1:0:0                  | 145k         | 128  | 50.0           | 86.8         | 15.0            |
> | InternLM2-Math-Plus-7B | N/A          | 16   | 13.5           | 52.0         | 2.5             |
> | InternLM2-Math-Plus-7B | N/A          | 32   | 20.0           | 61.1         | 3.0             |
> | InternLM2-Math-Plus-7B | N/A          | 64   | 33.0           | 67.5         | 5.0             |
> | InternLM2-Math-Plus-7B | N/A          | 128  | 38.9           | 73.0         | 7.5             |
>
>
> In the table, the ratio represents `original:tac_aug:inf_aug`, where:
>
> - original denotes the informal-formal pairs from the translated Mathlib4 dataset,
>
> - tac_aug refers to the tactic-based augmented statements, and
>
> - inf_aug corresponds to the informal augmented statements.
>
>
> All groups share the same "original" set, and we ablate the augmented datasets to assess the effectiveness of our methods. As a result, each group contains a different size of training data. The base model is trained on each dataset separately, and the checkpoint with the best performance is selected for each group. The model with the 1:2:1 ratio and 580k dataset size is trained on NL→FL dataset, and the model with the 1:2:1 ratio and 1160k dataset size is trained on NL→FL and FL→NL datasets. Note that during the ablation test, we discovered a checkpoint with the 1:2:1 ratio (the main experimental setting in our paper) that outperforms the one originally selected in the paper. We have already updated the results in the new version of the paper using this checkpoint.
>
> As shown in the table, our augmentation methods effectively enhance model performance, particularly on more challenging datasets such as ProofNet. The model trained solely on the original informal-formal pairs (1:0:0) achieves a respectable accuracy of 50.0\% on the ProofNet-valid dataset, 86.8\% on the MiniF2F-test dataset, and 15.0\% on the Extract-Theorem dataset, which underscores the quality of the informalized Mathlib4 dataset.
>
> The introduction of augmented datasets significantly boosts model performance across all datasets. The model trained with a 1:1:1 ratio shows a notable improvement, achieving 68.3\% on ProofNet-valid, 91.4\% on MiniF2F-test, and 17.5\% on Extract-Theorem. The 1:2:0 ratio model, which includes more tactic-based augmented statements, further enhances performance, reaching 74.7\% on ProofNet-valid, 95.1\% on MiniF2F-test, and 19.0\% on Extract-Theorem.
> The model trained with 1:2:1 ratio on both NL→FL and FL→NL datasets outperforms the model trained with 1:2:1 ratio on only NL→FL datasets, with accuracies of 81.6\% on ProofNet-valid, 96.7\% on MiniF2F-test, and 23.5\% on Extract-Theorem. This result indicates that increasing the size of the training set, while maintaining a balanced augmentation ratio, leads to substantial improvements in model performance, especially on more challenging datasets.
>
> We have integrated the ablation study results, along with the updated main results for the Herald Translator, into the new version of our paper.

---

> ### Author Response · Authors · 2024-11-27
> **Response to Reviewer 5xV4 (2/2)**
>
> (cont.)
>
> The above are quantitative results on augmented statements. To provide an illustrative understanding of the quality of our informal augmented statements, we include examples of informal augmented statements here, including multilingual augmentation as a strategy. Please refer to the newly added section Appendix E.3 for more relevant examples.
>
> ```
> theorem sum_not_convergent'_tac : Tendsto (fun n => ∑ i in Finset.range n, 1/i) atTop atTop := by sorry
> ```
>
> Let $n$ be positive natural number. The infinite sum of the sequence $\frac{1}{n}$ is not convergent.
>
> This statement is augmented to:
>
> 1. (Logical Equivalence Rewriting) The infinite sum of the sequence $\frac{1}{n}$ is divergent, where $n$ is a positive natural number.
>
> 2. (Abstract Concept Substitution) Let $n \in \mathbb{N}^+$. The harmonic series diverges.
>
> 3. (Omission of Implicit Condition) The sequence `1/n` does not have a convergent infinite sum.
>
> 4. (Multi-linguistic Translation) Soit $n$ un nombre naturel positif. La somme infinie de la suite $\frac{1}{n}$ n'est pas convergente.
>
> **R W2:** **Comparision to Deepseek-Prover**
>
> To the best of our knowledge, DeepSeek-Prover-1.5-SFT and DeepSeek-Prover-1.5-RL are not designed for the statement formalization task. These models are specifically developed for automatic theorem proving, where the input consists of already formalized statements. In our attempts to adapt these models for statement formalization, we found that they exhibit very limited capability in this area.
>
> **R W3: Contamination concerns**
>
> Thank you for raising these concerns. We have carefully addressed the potential contamination issue in three stages: the selection of the base model, the preparation of the training dataset, and the customization of benchmarks.
>
> We selected DeepSeek-ProverV1.5-Base as our base model, which is further pre-trained from DeepSeekMath-Base [1]. This pre-training process involves filtering out specific web pages to avoid contamination with commonly used benchmarks. DeepSeek-ProverV1.5-Base is trained on high-quality datasets comprising both code and natural language mathematical content. Since its performance has been evaluated on MiniF2F-Test and ProofNet, it is unlikely that the model has been exposed to these datasets during training.
>
> The Herald dataset was constructed exclusively using data from Mathlib4, which does not contain problems from MiniF2F or ProofNet. The Herald Translator is fine-tuned using only the Herald dataset and OpenHermes2.5 (a natural language instruction fine-tuning dataset). Therefore, the likelihood of contamination during this step is minimal.
>
> Beyond the commonly used benchmarks MiniF2F and ProofNet, we additionally tested performance on two custom datasets: Extract Theorem and College CoT. Extract Theorem is a dataset compiled by extracting theorems from advanced undergraduate-level textbooks using OCR on scanned materials. College CoT is a curated dataset derived from digital mathematical resources found online, with content verified and filtered using a large language model (LLM) to ensure quality and relevance. Both datasets are derived from real-world formalization tasks and do not consider infrastructure completeness in Mathlib4, making them more challenging to formalize. Importantly, this is the first time the informal theorems in these two datasets have been formalized into formal theorems, and their creation does not involve content in Lean. This ensures there are no contamination issues with these benchmarks.
>
> Considering the careful data preparation procedures and the high performance of the Herald Translator on Extract Theorem and College CoT, we believe that contamination is not an issue in our experimental settings.
>
> [1] Shao, Zhihong, et al. "Deepseekmath: Pushing the limits of mathematical reasoning in open language models." _arXiv_ _preprint_ _arXiv:2402.03300_ (2024).

---

### Author Response · Authors · 2024-12-04
**General Response: Evaluation of the Dataset Quality (1/2)**

We sincerely thank all the reviewers for their insightful and high-quality feedback. We are glad to see that the reviewers have praised the retrieval-augmented generation pipeline for dataset creation. Additionally, we are gratified to see that the reviews acknowledge the high quality of our dataset, as evidenced by the provided examples, and the effectiveness of our translator, which extends beyond standard benchmarks to include real-world formalization projects.

In the following general response, we address some common questions raised in the reviews. Following the reviewers’ suggestions, additional experiments for evaluating the dataset quality have been included in the new version of the paper.

1. **How to conduct a convincing quantitative experiment to exhibit the Herald translator’s quality?** The reviewers suggest improvements to our quantitative experiment from three different aspects. Firstly, in terms of comparing with more base models, Reviewer 5xV4 recommends a more comprehensive comparison with DeepSeek-Prover-1.5 models [1]. Reviewer BoSn suggests comparing the results with FVEL [2] and FormalAlign [3]. Secondly, regarding the test set, Reviewer 6Lbv inquired about the quality of our test sets **Extract Theorem** and **CollegeCoT**. Reviewer 4Ljo suggests that we should also test on ProofNet. Lastly, in terms of metrics, Reviewer 6Lbv asks about the effect of the NLI check, and Reviewer 4Ljo requests more results of pass@k for different k values to better understand the capability of our model.

These are very good questions and suggestions that can enhance the persuasiveness of our experiments demonstrating the Herald translator’s ability. First of all, there are relatively few models specifically designed for the task of auto-formalization in Lean. In our attempts to adapt DeepSeek-Prover-1.5-SFT and DeepSeek-Prover-1.5-RL models for statement formalization, we found that they exhibit very limited capability in this area. Additionally, the auto-formalization methods in FVEL were implemented using Isabelle, which has different mathematical libraries compared to Lean. FormalAlign, which was recently released on arXiv (14 October 2024), focuses on detecting misalignments between informal and formal statements rather than performing auto-formalization tasks. To the best of our knowledge, TheoremLlama is the most closely related work addressing both the informalization of Mathlib4 and auto-formalization. We have compared our results against it both quantitatively and qualitatively.

For the test sets, to fully demonstrate the capabilities of our model, we prepared two datasets that contain a broad spectrum of advanced undergraduate-level material, including abstract algebra, partial differential equations, and stochastic processes, as well as more advanced graduate-level mathematical materials. In response to Reviewer 4Ljo’s suggestion, we have included results for ProofNet [4], a widely recognized benchmark in undergraduate-level mathematics. This addition allows for a more standardized and widely accepted evaluation of our model's performance.

For the metric, the reviewers’ insights are invaluable. Reviewer 4Ljo’s suggestion to include pass@k results for smaller k values highlighted our model’s robustness and efficiency, and we will incorporate this experiment in our subsequent version of the paper. Reviewer 6Lbv pointed out the issue that the NLI check may introduce false positive cases. We acknowledge that even the most stringent LLM-based NLI checks can yield false positives. For our application, the primary goal of the NLI checking pipeline is to ensure a fair performance comparison between models, as it remains one of the few practical approaches for large-scale automated evaluation. To address this, we conducted a human expert examination of the successful results from the language model NLI check, demonstrating that the false positive rates of the Herald translator and other models are comparable, thereby ensuring a fair performance test. We are excited by recent advancements in large-scale mathematical statement alignment, such as [3], and plan to incorporate these methods into future work.

[1] Shao, Zhihong, et al. "Deepseekmath: Pushing the limits of mathematical reasoning in open language models." arXiv preprint arXiv:2402.03300 (2024).
[2] Lin, Xiaohan, et al. "FVEL: Interactive Formal Verification Environment with Large Language Models via Theorem Proving." arXiv preprint arXiv:2406.14408 (2024).
[3] Lu, Jianqiao, et al. "FormalAlign: Automated Alignment Evaluation for Autoformalization." arXiv preprint arXiv:2410.10135 (2024).
[4] Azerbayev, Zhangir, Bartosz Piotrowski, and Jeremy Avigad. "ProofNet: A benchmark for autoformalizing and formally proving undergraduate-level mathematics problems." Second MATH-AI Workshop. 2022.

---

> ### Author Response · Authors · 2024-12-04
> **General Response: Evaluation of the Dataset Quality (2/2)**
>
> 2. **Ablation study to assess the effect of each component in the data generation pipeline** Reviewer 5xV4 and Reviewer BoSn both recommended conducting an ablation study to understand the contribution of each component in the data generation and augmentation process. This study will help identify which parts of the pipeline are most beneficial and guide future improvements.
>
> We appreciate the reviewers’ interest in the effectiveness of each part of the pipeline. To quantify the impact, we conducted an ablation study that evaluated the Herald translator's performance with varying amounts and proportions of synthesized statement pairs (generated through informal augmentation and tactic augmentation) during the training process. These findings have been included in the paper.
>
> However, focusing solely on the Herald translator's performance in the auto-formalization task might overlook the dataset's quality in other relevant tasks, such as formal-informal alignment, semantic embeddings, and its use as natural language annotations for Mathlib. We hope our dataset will be of high quality for all these tasks, so we conducted extensive qualitative case studies to demonstrate its comprehensive quality.
>
> Regarding the specific effects of each RAG method, due to the high cost associated with the LLM-based pipeline for dataset generation, we tested the impact of each RAG method on a small scale before generating the entire dataset. The effects of RAG methods on selected examples are detailed in the qualitative analysis of dataset examples in the appendix.

---

### Meta-Review · Area_Chair_EuJ7 · 2024-12-21

**Metareview:**

This paper presents a novel approach to synthesize parallel corpora of natural language statements and formal language proofs for math problems. A retrieval-based approach is used in the informalization process of translating formal proofs to natural language sentences. The author used the proposed method to create a natural language version of Mathlib4, and trained a transformer that formalizes natural language statements to Lean proofs. Extensive experiments are performed on both standard benchmark datasets as well as real-world in-the-wild mathematical literature.

**Strengths:** Reviewers found that the proposed method is “well designed” (6Lbv), “innovative” and “practical” (BoSn). The to-be-open-sourced tooling and data will be “helpful for future research in AI-assisted math” (5xV4), and have “clear use to the community” (4Ljo). The evaluation results are “impressive” since “they are beyond standard benchmarks” … “on a real-world formalization project” (5xV4), with “very high performance on translating miniF2F problems, and improved performance on more difficult college-level datasets” (4Ljo).

**Weaknesses:** There are issues around lack of certain ablations as well as potential issues with the LM NLI check and  the pass@K evaluation metric. Most of the issues are addressed during the rebuttal phase with some clarifications from the authors and additional experiment results. Some remaining issues are:

* 4Ljo: “clarity of statement vs proof autoformalization (which the authors have made clear in the above comments, but should still be amended in the paper)”

* BoSn: Ablation studies to analyze the impact of different components of our RAG system are impractical due to the size of input data (2.2B tokens).

However, none of these issues are major. Therefore, the decision is Accept.

**Additional Comments On Reviewer Discussion:**

Please see above.

---

### Decision · Program_Chairs · 2025-01-22

Accept (Poster)